



# Observations and the source investigations of boundary layer BrO in Ny-Ålesund Arctic

Yuhan Luo[1], Fuqi Si[1], Haijin Zhou[1], Ke Dou[1], Yi Liu[2] and Wenqing Liu[1]

[1]Anhui Institute of Optics and Fine Mechanics, Key Laboratory of Environmental Optics and Technology, Chinese Academy
of Sciences, Hefei, 230031, China

[2]National Synchrotron Radiation Laboratory, University of Science and Technology of China, Hefei, 230027, China

*Correspondence to*: Yuhan Luo (yhluo@aiofm.ac.cn) and Fuqi Si (sifuqi@aiofm.ac.cn)

**Abstract.** Bromine monoxide is a reactive halogen species which has crucial impact on the chemistry of the tropospheric polar boundary layer. During polar spring, BrO enhancement can be detected in both northern and

southern Polar Regions, while the boundary layer ozone depletion events occur. A considerable challenge for understanding enhanced BrO and the associated ODEs is the difficulty of real-time observations. In this study, a typical process of enhanced bromine and depleted ozone in late April, 2015 at Ny-Ålesund boundary layer was observed using ground-based Multi Axis-Differential Optical Absorption Spectroscopy (MAX-DOAS) technique. The results showed that there were as high as $8 \times 10^{14}$ molecular cm$^{-2}$ BrO slant columns above the Kings Bay area in 26 April. Considering

meteorology, sea ice distribution and air mass history, the floating sea ice in the Kings Bay area was considered as the major source of this bromine enhancement event. During this period, the boundary layer ozone and gaseous elemental mercury (GEM) was synchronously reduced by 85% and 90% separately. The kinetic calculation showed that the ozone loss rate is 10.3 ppbv h$^{-1}$, which is extremely high compared to other area. The GEM loss rate is about 0.25 ng m$^{-3}$ h$^{-1}$. The oxidized GEM may directly deposit to snow/ice and thereby influence the polar ecosystem.

## 1 Introduction

Bromine monoxide is one of the key reactive halogen species and to have profound impacts on the atmosphere chemistry of the polar boundary layer (PBL), especially the oxidative capacity of the troposphere (Saiz-Lopez and von Glasow, 2012). According to satellite (Richter et al., 1998); (Platt and Wagner, 1998;Wagner et al., 2001;Sihler et al., 2013), aircraft (Neuman et al., 2010;Pöhler et al., 2013), ground-based (Frieß et al., 2004;Frieß et al., 2011;Hönninger

and Platt, 2002;Saiz-Lopez et al., 2007;Simpson et al., 2007a;Tuckermann et al., 1997), and ship-borne observations (Bottenheim et al., 2009;Jacobi et al., 2006;Leser et al., 2003;Wagner et al., 2007), BrO enhancement usually occurred over the frozen ocean during polar spring, while the boundary layer ozone dropped from typical levels (about 30 ppbv) to few ppbv, called "ozone depletion events" (ODEs). The duration of ODEs at coastal sites is typically between 1-3 days, depending on meteorology (Simpson et al., 2007b).

Bromide from sea salt aerosol is a sufficiently strong bromine source to explain the rate of ozone destruction or





observed BrO enhancements. Sea-ice (first year) surfaces, brine, and frost flowers have been considered as possible former of bromide aerosols (Kaleschke et al., 2004). The reactive bromine species (Br and BrO radicals) are released via a series of photochemical and heterogeneous reactions, known as the "bromine explosion" (Platt and Hönninger, 2003). Moreover, bromine-rich air masses can also be transported over land by monsoon or air turbulence. However,

since it is difficult to make detailed chemical observations in source area, direct evidence of bromine source involvement is sparse. One recent study has shown rapid ozone depletion (~7h) in the marginal sea ice zone, which has been interpreted as due to in-situ chemical loss (Jacobi et al., 2006).

Briefly, the chemical ozone depletion process is driven by the BrO cycling reaction in the PBL (Platt and Hönninger, 2003). A typical heterogeneous reaction model between gaseous and condensed phases was shown in Fig.1. Bromine is

released from salty ice surfaces to the atmosphere in an autocatalytic chemical mechanism that oxidizes bromide to reactive bromine. The reaction of HOBr in aerosol is proposed to be the pivot to explain the recycling reaction, which is an acid-catalyzed reaction (Simpson et al., 2007b).

To understanding these processes,optical remote sensing techniques are employed to detect BrO in the PBL relying on their characteristic narrow band absorption structures at UV and visible wavelengths. Ground-based multi-axis DOAS

(Differential Optical Absorption Spectrometer) technique has the advantage of being able to separate clearly the tropospheric and stratospheric portions of the atmospheric column, and even derive a crude vertical profile (Frieß et al., 2011). When pointing to a direction slightly above the horizon, high sensitivities for the trace gases close to the ground can be obtained due to the long light path through the trace gas layers. It is also an important calibration of satellite observations, which usually underestimates the tropospheric trace gases concentrations (Platt and Wagner, 1998).

In the Arctic area, ground-based observations have been made at Barrow, Alaska (71 °N, 157 °W), Alert, northern Canada (82.5 °N, 62.3 °W) and Ny-Ålesund, Svalbard (Tab.1). Long-path DOAS measurement provides regional determination of BrO in PBL, but few ground-based MAX-DOAS measurement of BrO has been performed in Ny-Ålesund. One of the reasons is that influenced by the North Atlantic Warm Current (NAWC), the near surface air temperature and sea surface temperature (SST) of East Greenland and North of Europe are relatively high (Fig.2).

The sea ice concentration has dropped severely during Arctic spring. Some areas even have no sea ice at all. In recent years, Kings Bay of Ny-Ålesund has ice-free open water all year round, which is an unique character comparing with other parts at the same latitude in Arctic. The modeling study indicated that sea salt surface (brine, dry salt, or possibly frost flowers) on sea ice is one of the most important prerequisites for a polar bromine explosion (Lehrer et al., 2004). Therefore, high level of troposphere BrO can be detected much more frequently in East Arctic (coastal area of north

Asia and North America) because of widely distributed sea ice.

In this study, we have discovered a typical process of enhanced bromine and depleted ozone in Ny-Ålesund in late April. The key role of bromine was confirmed by ground-based MAX-DOAS measurements of BrO. This event

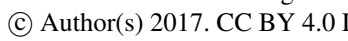


provides a rare opportunity to investigate the source of bromine and process of ozone depletion at this area. Furthermore, the mechanisms and environment implications of ozone depletion and gaseous mercury deposition are discussed.

**2 Instruments and methods**

**2.1 Instrument setup**

The MAX-DOAS measurement site is located at Yellow River Station (78°55′30″N, 11°55′20″E) at Ny-Ålesund, west coast of Spitsbergen. The observation position is shown in Fig. 3. To have a rough idea of the climate condition, monthly mean sea ice concentrations anomalies and air temperature anomalies in April 2015 are demonstrated in Fig. 2. The observations were carried out from 24 April to 15 May 2015. Due to the wavelength adjustment, no data is

available from 29 to 30 April.

The MAX-DOAS instrument operated at Ny-Ålesund consists of indoor and outdoor parts. The telescope receiving scattered sunlight from multi angles is controlled by a stepper motor to adjust elevation angles from horizon (0°) to zenith (90°). The field of view of the telescope is about 1°. The scattered sun light is imported through the quartz fiber with numerical aperture of 0.22 into the indoor spectrograph (Ocean Optics MAYA pro) with a one dimensional CCD

(ILX511 linear array CCD) containing 2048 pixels. The wavelength range of the spectrograph is from 290 nm to 420 nm, thus enabling the analysis of trace gases including $O_3$, $NO_2$, BrO, HCHO, and $O_4$. The spectral resolution is about 0.5nm (FWHM). The CCD detector is cooled at -30℃ while the whole spectrometer is thermally stabilized at +20℃ using a thermal controller. A computer sets the configuration of the system and controls the automatic measurements. The integration time of each measurement depends on the intensity of scattered light which can be influenced by cloud

and visibility. The standard mercury lamp is used for spectra calibration. Calibration measurements of dark current and offset are performed after each measurement.

The telescope is pointed towards Northeast direction, which covers the Kings Bay area (Fig.3). The sequence of elevation angles is 2°, 3°, 4°, 5°, 6°, 8°, 10°, 15°, 30° and 90° above the horizon.

**2.2 Data evaluation**

The spectra measured with the above described setup are analyzed using the well-established DOAS retrieving method (Platt, 1994). The wavelength calibration was performed using the QDOAS software developed by the Belgian Institute for Space Aeronomy (BIRA) by fitting the reference spectrum to a high resolution Fraunhofer spectrum (Kurucz et al., 1988). The spectral analysis of BrO is performed at 340-359 nm, encompassing three BrO absorptions bands, which improves the accuracy of the inversion. $O_3$ (223K, 243K) (Bogumil et al., 2003; Vandaele et al., 1998),





NO$_2$ (298K, 220K) (Vandaele et al., 1998), O$_4$ (Hermans et al., 2003), BrO (228K) (Wilmouth et al., 1999), OClO

(233K) (Kromminga et al., 2003), and Ring Structure (Chance and Spurr, 1997) are involved in the inversion algorithm.

Noontime zenith sky measurements from 26 April is chosen as Fraunhofer reference for the analysis. The O$_4$ retrieval

is performed using the same set of cross sections as for BrO but in the wavelength interval at 340-370 nm. The high

resolution cross sections were convoluted with the instrument slit function determined by measuring the emission line

of a mercury lamp. A fifth order of polynomial was applied to eliminate the broad band structures in the spectra caused

by Rayleigh and Mie scattering. Furthermore, a nonlinear intensity offset was included in the fit to account for possible

instrumental stray light. A wavelength shift and stretch of the spectra was allowed in the fit in order to compensate for

small changes in the spectral adjustment of the spectrograph.

The fit procedure yields differential slant column densities (dSCD) for the target gas. An example of the fit result of

BrO and O$_4$ is shown in Fig.4. The spectrum was recorded on 26 April, 2015 19:59 UTC (SZA=86 °) at the elevation

angle of 2 °. The BrO dSCD is $5.10 \times 10^{14}$ molecular cm$^{-2}$. The residual root mean square is $4.59 \times 10^{-4}$, resulting in a

statistical BrO dSCD error of $1.63 \times 10^{13}$ molecular cm$^{-2}$.

Since SCD is dependent on the light path, wavelength and observation geometry, SCD is then converted to vertical

column density (VCD) by dividing the air mass factor (AMF), which is the averaged light path enhancement for solar

light traveling through the atmosphere compared to a straight vertical path.

According to Sinreich et al (Sinreich et al., 2013), the effective light path length (*L*) in the boundary layer can be

determined by the O$_4$ dSCD for low elevation angle. The diurnal variations of the clear-sky AMFs for BrO are in

accordance with O$_4$ in the boundary layer.

$$L = \frac{dSCD_{O4}}{c_{O4}}$$
         (Eq. 1)

The near-surface O$_4$ concentration ($c_{O4}$) is calculated to be $2.8 \times 10^{37}$ molecules$^2$cm$^{-6}$ using local atmospheric

temperature and air pressure, which is proportional to the square of the O$_2$ concentration.

Considering the wavelength dependence of Rayleigh and aerosol scattering, the effective light path systematically

varies with the wavelength. Since the light path length based on O$_4$ dSCD is obtained at 360 nm, extrapolating *L* to

other wavelength ranges for retrieving trace gases such as BrO (350 nm) is an important aspect. The relationships

between the light path lengths at 350 nm to those at 360 nm were studied using SCIATRAN for a variety of aerosol

scenarios. The effective light path (*L*$_{eff}$) can be approximately calculated as:

$$L_{eff} = -0.017 + 1.003 \times L - 0.011 \times (L)^2$$
         (Eq. 2)

Thereby BrO SCDs can be converted to near-surface volume mixing ratios (*VMR*) by dividing effective light path

(*L*$_{eff}$):

$$VMR = \frac{dSCD}{L_{eff}}$$
         (Eq. 3)





### 2.3 Complementary data

Ny-Ålesund is a science community hosting over fifteen permanent research stations. Atmospheric measurements have been measured continuously at Zeppelin Station, Ny-Ålesund since 1990. Located on Zeppelin Mountain with altitude 474 m a.s.l., it is a background atmosphere observatory operated by NPI (Norwegian Polar Institute) and NILU

(Norwegian Institute for Air Research) , which are part of the Global Atmosphere Watch (GAW) Framework.

Surface ozone was measured by UV photometry. Gaseous mercury in air was measured using Tekran mercury detector. Hourly Surface Ozone and gaseous mercury data are downloaded at EBAS database (Tørseth et al., 2012).

Meteorology data including temperature, air pressure, relative humidity, wind direction and velocity, and global radiation are recorded by AWIPEV Atmospheric Observatory in Ny-Ålesund.

Webcam on the 474m Zeppelin Mountain records the sea ice change of Kings Bay and the cloud situation of Ny-Ålesund. (https://data.npolar.no/_file/zeppelin/camera/)

In order to get a rough idea of BrO distribution, station overpass BrO vertical column densities for MetOp-A (GOME-2A) and MetOp-B (GOME-2B) in Ny-Ålesund, Arctic are downloaded from https://avdc.gsfc.nasa.gov.

Using Hybrid Single-Particle Lagrangian Integrated Trajectory (HYSPLIT) model via the NASA ARL READY

website (http://www.ready.noaa.gov/ready/open/hysplit4.html) (Draxler and Rolph, 2013;Stein et al., 2015), back trajectory analyses were carried out to find the history of air masses. 72-hours ensemble back trajectories were driven by meteorological fields from the NCEP Global Data Assimilation System (GDAS) model output.

### 3 Results

Time series of BrO dSCDs at 2 °, surface ozone concentrations (from GAW station, Zeppelin), solar zenith angle (SZA),

air pressure, air temperature, relative humidity, wind velocity and wind direction from 24 April to 15 May are presented in Fig.5. Owing to the increase in stratospheric air mass factor at twilight, U-shaped diurnal variation of the BrO dSCDs are derived when BrO is mostly located in the stratosphere. From 26 to 28 April (high light blue area in Fig.5), BrO dSCDs clearly exceeded the background levels and peaked at $8 \times 10^{14}$ molecular $cm^{-2}$ in the evening of 26 April. At the same period, severe ODEs occurred with surface ozone sharply decreasing to several ppb and not

recovered until BrO had gradually dropped to background levels in 28 April. The occurrence of depleted troposphere ozone and enhanced BrO appears to be unpredictable in May. Partial ozone (not to near zero level) was depleted in the absence of BrO.

Fig. 6 provides the BrO variation from 25 April to 28 April in detail. Sunshine duration, global radiation, SZA, BrO dSCDs of 2 ° elevation angle, effective light path, BrO VMR, GOME-2 overpass BrO VCDs, retrieved BrO vertical

distribution, surface ozone and gaseous mercury are plotted. Light path derived from $O_4$ dSCDs (Fig. 6d) is about 18



km under cloud-free conditions, much longer than LP-DOAS (1-3 km) in previous researches (Frieß et al., 2011). The light path increases to more than 22 km in 27 April midnight due to the presence of cloud, which is consistent with the global radiation. The BrO VMR is about 15 pptv during the ODE (Fig. 6e).

BrO VCD from GOME-2 overpass data (Fig. 6f) shows that no obvious BrO enhancement has been detected, which

indicates that BrO at Arctic BL is underestimated by satellite-based measurements. Thereby, ground-based observations in this study are crucial complements, which provide calibration basis for the satellite observations.

In order to investigate the dynamical and chemical processes affecting bromine chemistry, BrO slant columns at different elevation angles are demonstrated in Fig. 7, which have contributed to better understanding of the vertical distribution of reactive bromine at Arctic boundary layer. The retrieval presents that BrO dSCDs are higher at low

elevation angles than those at high elevation angles. Furthermore, results of different elevation angles distinguish obviously, especially during the BrO enhancement period, which implies that BrO is generated from the sea surface. As a consequence, there is sufficient reactive bromine present locally in the boundary layer to explain the extremely fast $O_3$ depletion and mercury precipitation simultaneously (Fig. 6h, 6i).

## 4 Discussions

In this research, high concentration of troposphere BrO has been detected using the ground-based MAX-DOAS technique. As high as $8\times10^{14}$ molecular $cm^{-2}$ BrO column has been detected above Kings Bay, Ny-Ålesund. The retrieval shows that the enhancement occurred accompanied with severe ozone depletion and mercury precipitations. The maximum BrO is located at sea surface during the ODE. This enhancement event is a good opportunity to investigate the source of BrO and the impact on the environment of Arctic boundary layer. The following parts are

discussed in detail from air mass history, sea ice distribution, and ozone loss and mercury precipitation.

### 4.1 History of air masses

From the map of GOME-2 measurements from 22 April to 27 April 2015 (Fig. 8), several troposphere BrO clouds existed at coastal North America and Chukchi Sea on 22 April 2015 and gradually diffused. But from GOME2 overpass data at Ny-Ålesund, no apparent enhancement of BrO has been detected on 26 April. Moreover, long-range

transport of air masses (Fig. 9) did not show significant change during the fast ODE period. 72-hour backward trajectories at Ny-Ålesund 10 and 3000 meters a.s.l. from 26 April (0600 UTC) to 27 April (1800 UTC) were analyzed. The trajectories followed similar pathways, which indicate a stable circulation pattern for the period before, during, and after the BrO enhancement. Therefore it means that the observed BrO enhancement event is not due to the long-range transport of BrO precursors.





### 4.2 Sea ice distribution

Sea water is the primary source of the boundary layer halogens (Mcconnell et al., 1992). The sea ices provide highly concentrated saline surfaces, which have been reported to play a pivotal role in for bromine activation (Saiz-Lopez and von Glasow, 2012). According to the observation of sea ice concentration from AMSR-E and Zeppelin webcam, Kings

Bay is ice-free water area during the measurement period (Fig. 2). However, large amount of sea ice appeared at Kings Bay on 26 April (Fig.10), floating from the open water by both wind and tidal effect.

The efficient ozone loss is consistent with the temperature decline (Fig.11). The meteorology data shows that on 26 April air temperature continually goes down and reaches bottom of -11.4$^o$C at 22:00. According to the precipitation curve of calcium carbonate, large amount of carbonate precipitates below 265K. This process will provide acid aerosol

from alkaline sea water, which triggers the transformation of inert sea-salt bromide to reactive bromine (Sander et al., 2006). Although the sun radiation intensity is not strong at that time, the heterogeneous reactions can still happen under the twilight.

Thereby, this BrO enhancement event is a local process, mainly influenced by underlying surface change and local environment. The surface ozone concentrations increased along with the melting of sea ice, which indicated that the

life span of BrO radicals are very short. When sea ice disappeared, the reaction immediately ended and reactive bromine radicals gradually transformed to soluble bromide (e.g. HOBr), which explained the sink of it (Fan and Jacob, 1992).

### 4.3 Kinetic analysis of $O_3$ loss

What makes this case very special is that the increasing rate of BrO and the depletion rate of boundary layer ozone are

really fast. The surface ozone reduced by 85% within 4 hours. The ozone loss rate is as high as 10.3 ppbv h$^{-1}$ or 248 ppbv d$^{-1}$, which is extremely high compared with previous studies in Polar Regions (Tab. 2).

The chemical kinetics of bromine enhancement and ozone decay are analyzed assuming that the catalytic reactions are dominated by reactions showed in Fig.1. A first-order loss of ozone is due to reaction Br+$O_3$→BrO+$O_2$ resulting in the rate law:

$$r = -\frac{d[O_3]}{dt} = k_1 \cdot [O_3] \tag{Eq. 4}$$

$$[O_3] = [O_3]_0 \cdot \exp(-k_1 \cdot t) \tag{Eq. 5}$$

$$\ln\frac{[O_3]}{[O_3]_0} = -k_1 \cdot t \tag{Eq. 6}$$

$[O_3]_0$ is the ozone concentration at the beginning of decay determined from the measured mixing ratio of 74.72 ppbv.

$\ln\frac{[O_3]}{[O_3]_0}$ versus time are showed as hollow square in Fig. 12a.



According to the method by Jacobi et al. (Jacobi et al., 2006), the first order rate constant $k_1$ can be determined as follows:

$$\frac{d(\ln\frac{[O_3]}{[O_3]_0})}{dt} = -k_1 \qquad \text{(Eq. 7)}$$

The measured decrease of $\ln\frac{[O_3]}{[O_3]_0}$ versus time was fitted by:

$$\ln\frac{[O_3]}{[O_3]_0} = -\exp(b \cdot t + a) \qquad \text{(Eq. 8)}$$

$$\frac{d(\ln\frac{[O_3]}{[O_3]_0})}{dt} = -b \cdot \exp(b \cdot t + a) \qquad \text{(Eq. 9)}$$

$$\ln(-\ln\frac{[O_3]}{[O_3]_0}) = b \cdot t + a \qquad \text{(Eq. 10)}$$

$$k_1 = b \cdot \exp(b \cdot t + a) \qquad \text{(Eq. 11)}$$

$\ln(-\ln\frac{[O_3]}{[O_3]_0})$ versus time are plotted as black dots in Fig. 12a. The coefficients a and b are obtained from the linear fit in plot.

The ozone loss begins relatively slow and accelerates with time, which is consistent with the process of bromine explosion.

Assuming that the first order decay is dominated by the reaction Br+$O_3$→BrO+$O_2$, we are able to calculate the Br concentrations as follows:

$$k_1 = k_{Br} \cdot [Br] \qquad \text{(Eq. 12)}$$

$$k_{Br} = 1.7 \cdot 10^{-11} \cdot \exp(-\frac{800}{T}) \qquad \text{(Eq. 13)}$$

$k_{Br}$ is a constant depending on temperature (Fig. 12b). Thereby, the calculated Br concentration increases from $1.1 \times 10^7$ to about $1.2 \times 10^9$ atoms cm$^{-3}$ (corresponding to 44.8 pptv) (Fig. 12c). Considering the assumption that the halogens are homogenously distributed in the PBL, the concentrations of Br at sea surface layer in the bromine explosion could be even higher.

### 4.4 Gaseous mercury precipitation

During ODEs, halogens efficiently oxidize gas-phase mercury and cause it to be transferred from the atmosphere to the snow, called "atmospheric mercury depletion events (AMDEs)". Since 1995, year-round monitoring of atmospheric Hg has been performed at Alert, Canada (Schroeder et al., 1998). Dynamic studies demonstrated that bromine is the major oxidant depleting Hg in the atmosphere (Ariya et al., 2002;Ariya et al., 2004;Lindberg et al., 2002;Lu et al., 2001;Steffen et al., 2008).

In this case, the precipitation of gaseous mercury occurred concurrently with tropospheric ozone depletion, as well as the enhancement of BrO (Fig. 13), which indicated that the oxidation of GEM by reactive halogen species (Br atoms and




BrO radicals) is considered to be the key process of mercury depletion. The GEM decreases from ~2 ng m$^{-3}$ to lower than 0.3 ng m$^{-3}$ during the BrO enhancement event. The mercury loss rate is about ~0.25 ng m$^{-3}$ h$^{-1}$ or 6 ng m$^{-3}$ d$^{-1}$. The oxidized GEM may directly deposit to snow/ice or associate with particles in the air that can subsequently deposit onto the snow and ice surfaces, and thereby threaten polar ecosystems and human health.

**5 Conclusions**

Typical process of enhanced bromine and depleted ozone in Ny-Ålesund boundary layer was observed using ground based MAX-DOAS techniques in late April, 2015. As high as $8 \times 10^{14}$ molecular cm$^{-2}$ BrO slant columns were detected on 26-27 April. Meanwhile, severe ozone depletion and mercury precipitation occurred under BrO VMR of 15 pptv. The distribution showed higher BrO concentrations near sea surface. By analyzing the air mass history and sea ice

conditions, this BrO enhancement is a local process. The underlying sea ice and low temperature provide acid aerosols, which are prerequisites for the formation of BrO radicals. The kinetic analysis shows that the ozone loss begins relatively slow and accelerates with time, which is consistent with the process of bromine explosion. The ozone loss rate is as high as 10.3 ppbv h$^{-1}$, which is much higher than previous studies in Polar Regions. GEM loss rate is about ~0.25 ng m$^{-3}$ h$^{-1}$. This study is a pivotal complement for BrO research in Arctic BL. Further observations and analysis

are required to identify the chemical mechanisms.

*Acknowledgements.*

This research was financially supported by the National Natural Science Foundation of China Project No. 41676184, 41306199 and U1407135. We gratefully thank the Chinese Antarctic and Arctic Administration and the teammates of 2015

Chinese Arctic Expedition. We are also grateful to Dr. Ping Wang from KNMI and Dr. Yang Wang from MPIC for providing the advice on BrO VMR calculation. We kindly acknowledge the AWIPEV Atmospheric Observatory in Ny-Ålesund, the Norwegian Polar Institute and Norwegian Institute for Air Research (NILU) for the complementary data. Caroline Fayt, Thomas Danckaert and Michel van Roozendael from BIRA are gratefully acknowledged for providing the QDOAS analysis software. Meteorological data, surface ozone, and gaseous mercury were provided by EBAS database. Back trajectories were

calculated using the HYSPLIT model from NOAA together with the GDAS data set from NCEP.

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



**Table 1. Comparisons of BrO mixing ratio at four main Arctic observation sites**

| Sites | Observation periods | BrO mixing ratio | Methods | References |
|---|---|---|---|---|
| Greenland ice sheet (72N, 38W, 3200ma.s.l.) | 14 May-15 June 2007, 9 June-8 July 2008 | 3-5ppt | LP-DOAS | (Stutz et al., 2011) |
| Barrow, Alaska (71°19'N, 156°40'W) | 26 February-16 April 2009 | ~30ppt | MAX-DOAS LP-DOAS | (Frieß et al., 2011) |
| Alert, Nunavut (82°32'N, 62°43'W) | 20 April- 9 May 2000 | ~30ppt | MAX-DOAS | (Hönninger and Platt, 2002) |
| Ny-Ålesund, Svalbard (78.9N, 11.8E) | 20 April-27 April 1996 | ~30ppt | LP-DOAS | (Tuckermann et al., 1997) |

**Table 2. Comparisons of BrO mixing ratio and ozone loss rate**

| Method | BrO mixing ratio | Typ. Rate of O3 destruction | References |
|---|---|---|---|
| Observation at PBL | up to 30 pptv | 1-2 ppbv h-1 | (Tuckermann et al., 1997;Hönninger and Platt, 2002) |
| Observation at MIZ | ~63 pptv | 6.7 ppbv h-1 or 160 ppbv d-1 | (Jacobi et al., 2006) |
| Observation at salt lakes | up to 176 pptv | 10-20 ppbv h-1 | (Hebestreit et al., 1999;Stutz et al., 2011) |
| Observation at Marine BL | ~2 pptv | ~0.05 ppbv h-1 | (Leser et al., 2003) |
| Model | 30-40 pptv | 7.6 ppbv d-1 | (Lehrer et al., 2004) |
| Model | 100 pptv | 40ppbv d-1 | (Fan and Jacob, 1992) |
| Observation at Ny-Ålesund BL | ~15 pptv | 10.3 ppbv $h^{-1}$ or 248 ppbv $d^{-1}$ | **this study** |

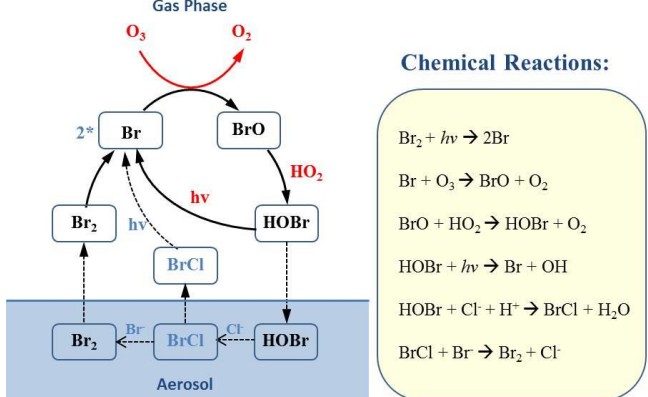

Fig. 1 Chemical reactions of BrO-Ozone cycle

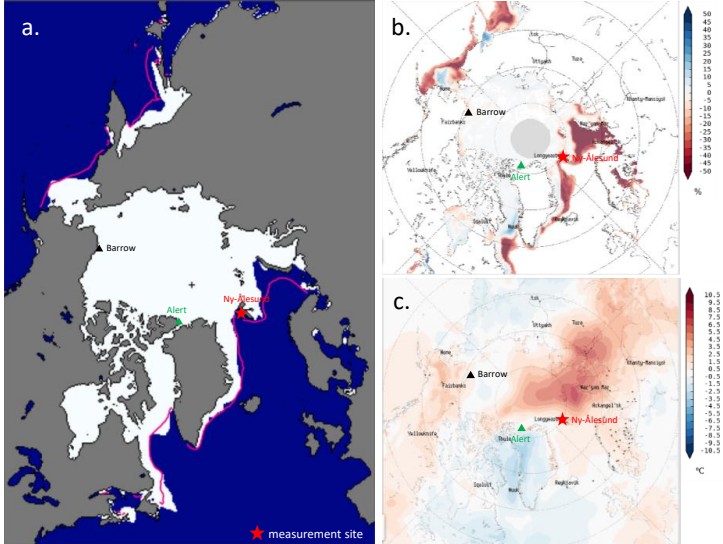

5     Fig 2. a. Sea ice extent of Apr 2015 in Arctic area (data from http://nsidc.org/data/seaice_index/); b. Monthly mean sea ice concentrations anomalies; c. Two meters air temperature anomalies of April 2015 compare to averages from 1979 to 2015 (data from http://nsidc.org/soac)





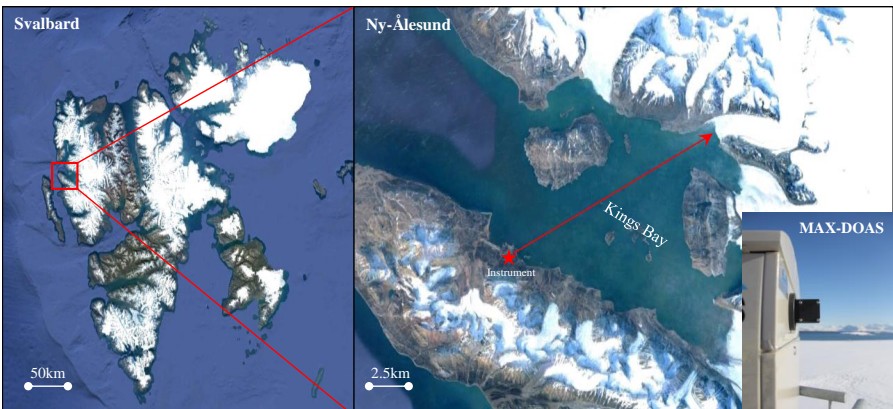

**Fig.3 MAX-DOAS field observation in Ny-Ålesund, Arctic**

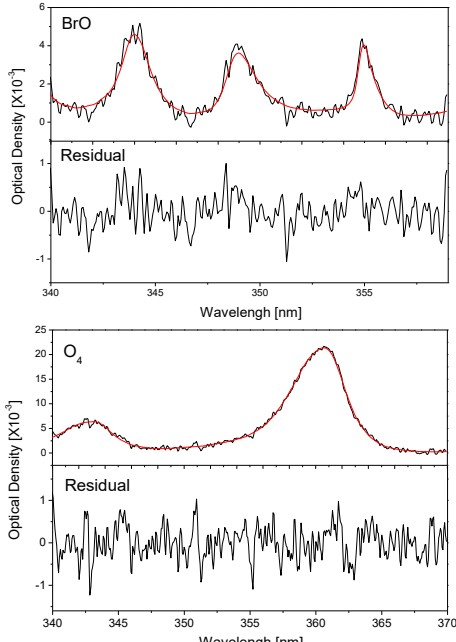

5    **Fig.4 Examples for spectral retrieval of BrO and O4. The spectrum was recorded under clear sky conditions at 2° elevation on 26 April 2015, 19:59 UTC, SZA = 86°. (Black lines: Retrieved spectral signatures fitted result for absorber; red lines: fitted cross sections)**



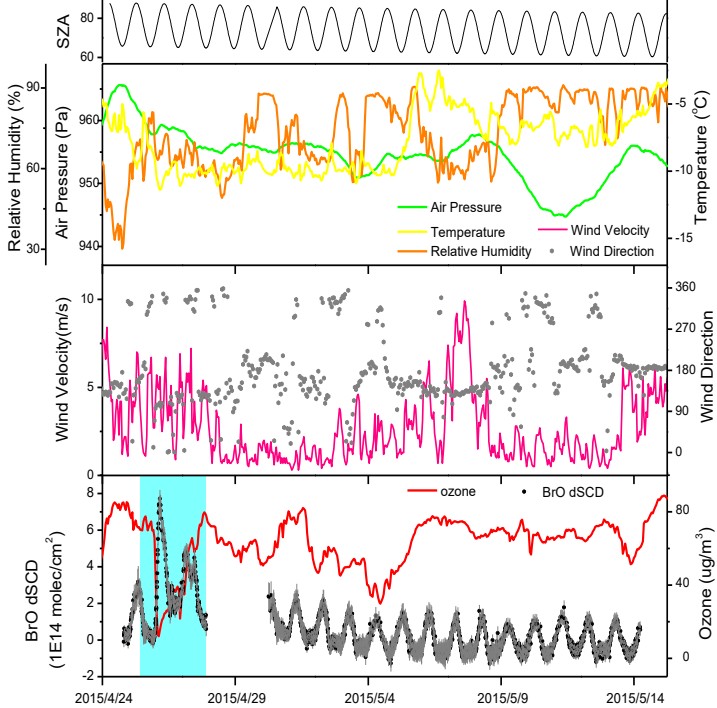

**Fig.5 Time series of BrO dSCDs, surface ozone, SZA and meteorology data during the measurement.**

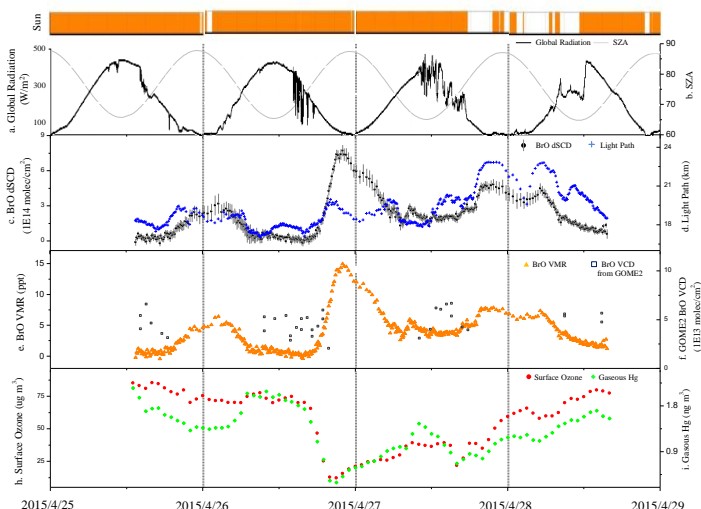

5    **Fig.6 a. Global radiation; b. SZA; c. BrO dSCDs from MAX-DOAS; d. Light path from MAX-DOAS; e. BrO VMR; f. BrO VCD from GOME2; h. surface ozone; and i. gaseous mercury during BrO enhancement event**



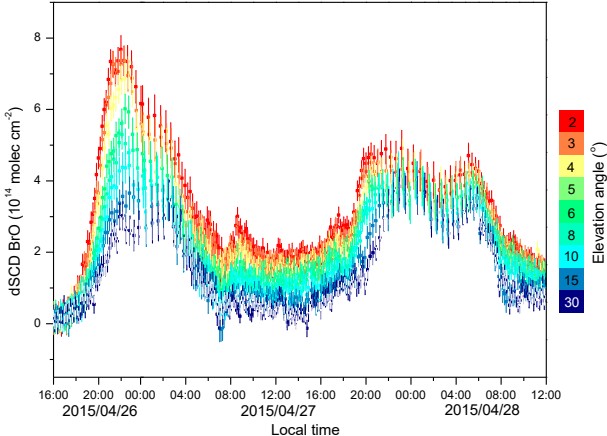

**Fig.7 BrO dSCDs of different elevation angles during the enhancement period**

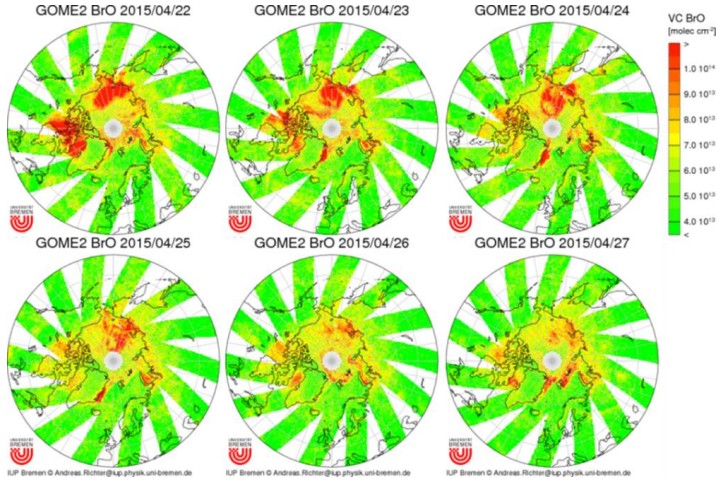

5    **Fig.8 Map of troposphere BrO of northern hemisphere by GOME-2 product from 22 to 27 April**

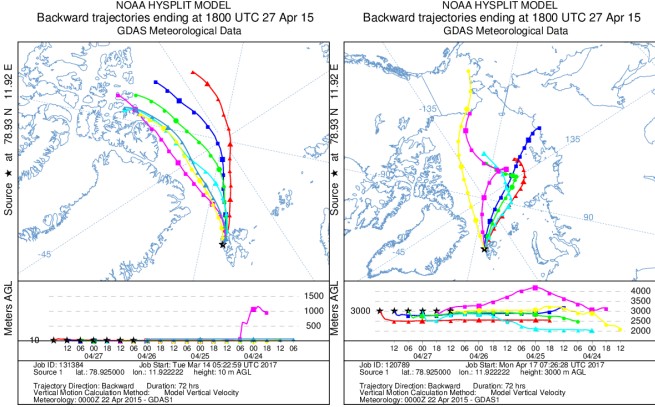

**Fig. 9 Back trajectory model of air masses arriving at Ny-Ålesund from 26 April (0600 UTC) to 27 April (1800 UTC) at two different altitude (10 meters and 3000 meters a.s.l.). Every 6h a new trajectory starts, each trajectory runs 72h.**





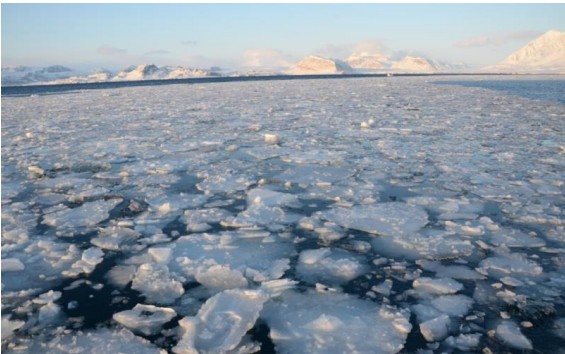

**Fig.10 Sea ice in Kings Bay, Ny-Ålesund at 22:00, 26 April 2015**

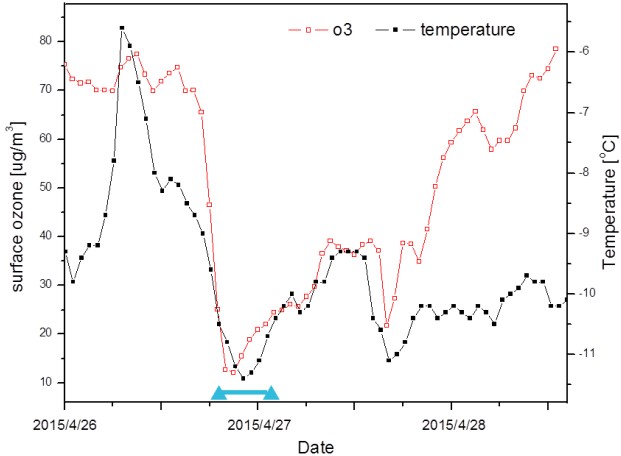

5    **Fig.11 Time series of surface ozone and air temperature during the BrO enhancement event, blue triangles present the sea ice existence in Kings Bay**





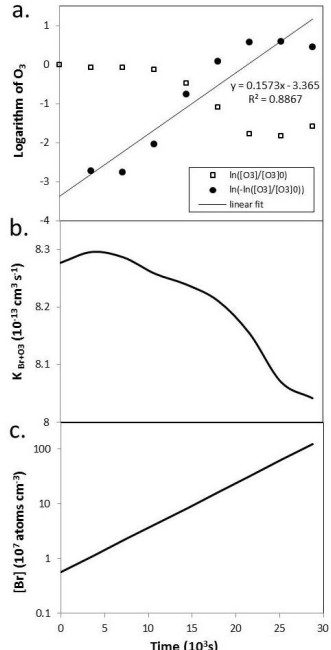

**Fig.12 Analysis of surface ozone loss in 26 April 2015**

**a. Plot of $\ln([O_3]/[O_3]_0)$ and $\ln(-\ln([O_3]/[O_3]_0))$ versus time; b. Calculated temperature dependent reaction rate coefficients for $O_3+Br$; c. Calculated Br concentration.**

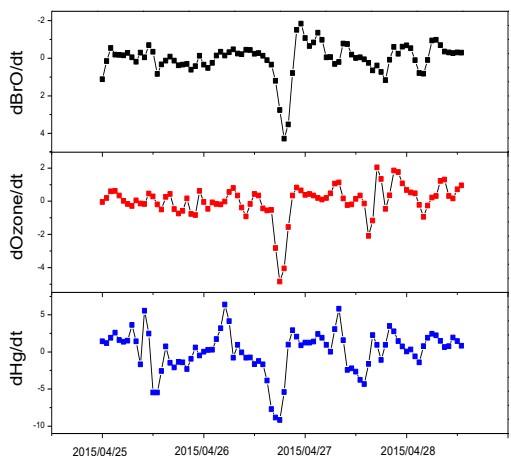

**Fig.13 Time series of dBrO/dt, dO3/dt and dHg/dt during the BrO enhancement event**