# Peer review of "Observations and the source investigations of boundary layer BrO in Ny-Ålesund Arctic"

_Atmospheric Chemistry and Physics, 2017_

## Referee Comment (RC1) · Anonymous Referee #1 · 8 Sep 2017

The paper describes an interesting event of enhanced tropospheric BrO concentrations at Spitsbergen. During that event very rapid decrease of O3 and mercury was observed.

In my opinion this is a very interesting case study and should finally be published in ACP. However, in its current form, the paper has a few major and many minor problems. They need to be addressed before acceptance.

Major problems

A) The introduction is not very clear. Please put your study in a better context of existing studies. For example, please make the following points more clear: - what are important current open research questions - what does this study contribute to answer

these questions

B) The discussion of the role of long range transport is not clear. In my opinion, from the data you present (trajectory and satellite data), long range transport of air masses enriched in bromine can not be excluded. This should be clearly stated in the paper. Nevertheless, from other findings, I think you can very well conclude that this is probable a local event: -the sea ice occurred only for a very short period close to the measurements -exactly during this period, the enhanced BrO is observed Both findings indicate that the enhanced BrO is caused by a local source (connected to the sea ice). Here, it would be very important to have more information: -how large was the area in which the sea ice appeared? -how does this area compare to spatial scale determined by the wind speed (and direction) and the duration of the ozone destruction? -how long have the air masses been in contact with the sea ice before they reached the measurement site? -I think it is in general very important here to discuss the importance of transport compared to chemical processes.

C) You use a fixed Fraunhofer reference spectrum. In my opinion this is not a good strategy here, because you are interested in tropospheric BrO DSCDs. I strongly suggest to use a sequential Fraunhofer reference. The QDOAS software which you use offers this option. Alternatively, you could subtract the BrO DSCDs from the 90° measurement of each elevation sequence from the BrO DSCDs at low elevation angles.

D) You apply the method of Sinreich et al. (2013). However, this method should be applied to the tropospheric DSCDs, for which the stratospheric part was already removed (both BrO and O4), see the point above. In the current form, the derived O4 light paths include the light paths in the free troposphere and stratosphere, which are misleading for your study.

E) From the dependence of the BrO DSCDs on elevation angle you conclude that the highest BrO concentration is located at the surface. If this was the case, then the differences of the BrO DSCDs for the low elevation angles should be much larger. In my

opinion, the highest BrO concentrations are probably not located directly at the surface. To clarify this important point I suggest to perform one of the following tasks: a) perform RTM simulations (for different assumed BrO profiles) and compare the RTM results to your measurement results. Then you can derive more robust conclusions about the BrO profile shape. b) it would be even better to perform a full profile inversion.

Minor points:

Page 1, line 24: Enhanced BrO was first detected by LP-DOAS observations. Please add the respective references.

Page 2, line 4: I think Monsoon and turbulence are not the correct terms here. Monsoon is a tropical to mid latitude phenomenon. Why not simply write 'wind'?

Page 2, line 19: I don't agree with this statement for observations over bright surfaces like ice or snow. I think for such conditions a general underestimation was not reported.

Page 2, line 21: You write 'Long-path DOAS measurement provides regional determination of BrO in PBL.' It is not clear what exactly you want to say with this. LP-DOAS usually has light path lengths between a few and 20 km.

Page 2, line 22: you write 'but few ground-based MAX-DOAS measurement of BrO has been performed in Ny-Ålesund.' I think you can not write this. To my knowledge, University of Bremen performs MAX-DOAS measurements there since many years. (Did you compare your results with the Bremen results?)

Page 3, line 9: Did you perform MAX-DOAS measurements also before and after the presented period? Have you observed similar events before or after, or in other years?

Page 3, line 9: What do you mean with 'wavelength adjustment'? Why do you have no measurements during such periods of wavelength adjustment?

Page 3, line 19: Please give typical integration times

Page 4, line 17: please add information about typical uncertainties of this method,

which are given in Sinreich et al., 2013. You might also have a look at the paper by Wang et al., 2015: Wang, Y., Li, A., Xie, P. H., Wagner, T., Chen, H., Liu, W. Q., and Liu, J. G.: A rapid method to derive horizontal distributions of trace gases and aerosols near the surface using multi-axis differential optical absorption spectroscopy, Atmos. Meas. Tech., 7, 1663-1680, https://doi.org/10.5194/amt-7-1663-2014, 2014.

Page 4, line 20: To my knowledge, the formula is only valid for the tropospheric DSCDs. That means that a) the DSCD for 90° elevation of the individual elevation sequence have to be subtracted from the DSCDs of the low elevation angles (for analysis with fixed reference) before the formula is applied b) or the analysis has to be performed with a sequential reference. Since you use QDOAS, I strongly recommend to use this option.

Page 4, line 24: In my opinion it makes no sense to apply such a correction, because of two reasons: a) the differences of the light paths for such small wavelength differences are very small b) the errors of this method (Sinreich et al) are generally rather large. Thus the uncertainties caused by the different wavelengths are negligible. I suggest to remove this part (lines 23 to 31)

Page 5, line 25: The results in Fig. 5 indicate that enhanced BrO (above the background) is found until 4 May 2015.

Page 5, line 29: I suggest to remove the GOME-2 BrO VCDs from the figure. You show total BrO VCDs. It is unclear how they relate to the BrO mixing ratios from the MAX-DOAS. If you want to keep the GOME-2 BrO VCDs, then you should discuss, how large the stratospheric BrO VCD is. And you should indicate, above which value of the total BrO VCD you think they are affected by enhanced tropospheric BrO.

Page 6, line 1: Since you use a fixed Fraunhofer reference, these light path lengths include the light paths in the free troposphere and stratosphere. This makes no sense here. I strongly suggest that you should use a sequential Fraunhofer reference (see major point above).

Page 6, line 4: at least on 27 April a slight enhancement is seen.

Page 6, line 9: The differences of the BrO DSCDs between $2°$ and $4°$ are very small (5 to 10%). In my opinion this indicates that the highest values are probably not directly above the surface. I suggest that you should perform RTM simulations (for different assumed BrO profiles) and compare them to the measurement results. Then you can derive more robust conclusions about the BrO profile. Alternatively also a full profile inversion could be done.

Page 6, line 12: how high is the boundary layer? Maybe the maximum BrO concentration is on the upper edge of the boundary layer? (see e.g. Wagner et al., 2007)

Page 6, line 28: In my opinion this can not be concluded. The trajectories pass over several regions with enhanced BrO (e.g. directly north of Spitsbergen). So, in principle air masses enriched in BrO might have been transported to Spitsbergen.

Page 7, line 15: The enhancement goes very slowly back to normal values (until 4 May). You might mentioned the period here.

Fig. 2b: to which reference (time period) are the anomalies calculated? Please give a reference for the map.

Technical suggestions / language improvement (please note that language improvement is urgently needed; here I list only a few points)

Page 1, line 11: 'difficulty of real-time observations' What exactly is meant here?

Page 1, line 17: separately => respectively?

Page 2, line 4: former => source?

Page 2, line 9: If you took this scheme from another publication, please add the corresponding reference.

Page 4, line 18: The sentence 'The diurnal variations of the clear-sky AMFs for BrO

are in accordance with O4 in the boundary layer.' is not clear to me. Please clarify.

Page 5, line 13: Are you sure that the BrO maps are from NASA? In Fig. 8 you show results from Uni Bremen.

Page 5, line 26: unpredictable => unexpected?

Fig. 5: It should be mentioned that presented BrO DSCDs are obtained from measurements at 2° elevation angle.

Fig. 8: Not the tropospheric, but the total BrO VCDs are shown. Please correct the caption. Would it be possible to show additional BrO maps for the period after 27 April?

---

## Referee Comment (RC2) · Anonymous Referee #2 · 20 Sep 2017

Ozone and mercury depletion events are recurrent phenomena characterizing the atmospheric chemistry of both polar regions during springtime. Despite more than 30 years of studies of such events, major gaps still exist in our knowledge on processes, necessary conditions, and amplitude of such events. One of the major difficulties is related to the fact that observed decreases in ozone and mercury concentrations can be caused by local chemical processes as well as the advection of already depleted air masses. Luo et al. report here new measurements of BrO concentrations performed at Ny-Alesund during a period of approximately three weeks in late April/early May 2015 with an episode of elevated BrO, which is one of the reactive species involved in the chemical destruction of ozone and the oxidation of mercury. Observed ozone and mercury concentrations show strong decreases parallel to the enhancement of BrO. After

evaluating meteorological conditions and remote sensing data, the authors propose that this is one of the rare cases with in situ observations of local processes leading to the depletion of ozone and mercury. While such a conclusion appears possible, I'm less convinced than the authors that the available data and analysis allows a firm decision about the influence of local chemistry versus atmospheric transport. I suggest a more extensive discussion of the limitations of the available data and potential scenarios with the corresponding pros and cons, so that each reader can form her/his personal opinion. Below I describe in more detail my major concerns that should be discussed in a revised manuscript.

Major comments

The authors claim that according to the trajectories in Fig. 9 the increase of BrO cannot be explained by the transport since all air masses do have the same origin. Nevertheless, the trajectory close to sea level arriving at the beginning of the BrO episode (18:00; 26/04) travelled closest to the northern coast of Greenland. However, some of the other trajectories shown in Fig. 9 originated from areas close to the North Pole. I believe that this can indicate different origins of the air masses. Moreover, reanalysis data (e.g. NCEP) show that the synoptic situation on that day was characterized by a strong high above Greenland and a large, but weaker low over Siberia. As a result, it appears possible that air masses traveling close to Greenland and influenced by the high may have had different properties and composition compared to those stemming from the North Pole and related to the low. Even stronger differences are visible in the backward trajectories at 3000 m altitude. (By the way, it remains unclear why these trajectories are shown.) Therefore, the statement that the "trajectories followed similar pathways, which indicate a stable circulation pattern . . ." (page 6, line 27) appears to overly simplifying the meteorological conditions. A more detailed analysis of the mesoscale situation could confirm (or not) the hypothesis of the authors.

Moreover, the BrO map for 26/04 in Fig. 8 demonstrates enhanced concentrations close to Greenland and probably lower concentrations further north (but due to the

lack of data this remains speculative). It appears possible that the elevated concentrations are related to the transport of air masses enriched in BrO (or at least air masses enriched in BrO precursors) and already depleted in ozone and mercury. To my knowledge, the most exceptional case of transport-related changes in ozone was presented by Morin et al. (Geophys. Res.Lett., 32, L08809, doi:10.1029/2004GL022098) demonstrating that ozone concentrations can vary on the time scales of minutes due to transport.

The BrO map as well as the BrO VCD further manifests an additional counter-argument of local chemistry driving the ozone and mercury depletion: The Ny-Alesund area is not part of the area with elevated BrO concentrations. In previous studies claiming local chemical processes (e.g. Jacobi et al. 2006) the measurement sites were located at least close to the area with enhanced BrO. I understand that the authors claim that differences between the satellite and ground measurements of BrO can occur (page 6, line 4ff), but in my opinion such differences, their origin, and potential consequences should be discussed in more detail.

The observations of BrO, ozone, and mercury were not obtained at the same altitude. The authors claim that a large fraction of the enhanced BrO was located close to sea level. First, I agree with referee 1 that the observations in Fig. 7 rather seem to indicate that the highest BrO concentrations were actually at higher altitudes. This needs to be clarified. Second, ozone and mercury were measured at the Zeppelin Station and, thus, at around 480 m a.s.l.. Air masses at Zeppelin Station often represent tropospheric air from higher altitudes and are regularly decoupled from the boundary layer at Ny-Alesund. Thus, the authors need to demonstrate that during the observed event, such a decoupling between Ny-Alesund and Zeppelin did not occur. Corresponding vertical profiles of meteorological data should be available for example from the French-German AWIPEV Station. Third, the vertical extent of ODEs and elevated BrO can be constrained to only a few hundred meters (e.g. Friess et al., JGR 116, D00R04, doi: 10.1029/2011JD015938). Therefore, the authors need to demonstrate

why during this event the observations at sea level and the observations at higher elevations were directly linked. Finally, if the event was local and started at sea level I would expect a delayed response in ozone and mercury at 480 m altitude. However, the time series shown in Fig. 6 suggest either a concomitant increase in BrO and decrease in ozone and mercury or even an onset of the decrease in ozone and mercury before the increase of BrO. This should be clarified. Only if the authors can confirm that increase in BrO and the decrease in ozone occurred in the same air mass, a kinetic analysis as presented in chapter 4.3 is useful (see below).

The authors state that the sea ice shown in Fig. 10 suddenly appeared in the fjord (page 7, line 5f). They further propose that the sea ice is the source of the reactive BrO. This hypothesis seems to be based on previous studies claiming that freshly formed sea ice or first-year sea ice are major sources of reactive bromine. However, the sea ice shown in Fig. 10 does not resemble freshly formed ice. The photograph rather shows crushed pieces of ice stemming from older floes and submerged in sea water. It appears that this sea ice was not formed in the fjord, but rather transported into the fjord by wind and wave actions as mentioned by the authors. Why this type of sea ice should lead to a sustained formation of reactive halogens remains, thus, unclear. I'm also convinced that the claim of the authors that this sea ice type fosters the formation of acidic sea salt aerosols is not warranted (page 7; line 8ff). Although air temperatures are low, the temperature of the crushed ice floes is probably close to the temperature of the water in the fjord, which can only be as low as $\sim$-2$°$C. As a result the claimed precipitation of calcium carbonate supporting the acidification of the aerosols formed on the observed sea ice (or even sea water as claimed by the authors; page 7, line 9f) is not likely. In any case, a more detailed description of the ice conditions and how they developed during the days before the event would be necessary and useful.

The maximum Br concentrations derived from the kinetic analysis are higher than the BrO concentrations derived from the DOAS measurements ($\sim$45 pptV vs. 15 pptV). First of all, such a kinetic analysis can only be done if it is assured that the measurements concern the same air masses, which is not obvious with the different altitude of the observations (see above). In any case, is this a realistic result? Is the calculated Br too high or the observed BrO too low? This can also be interpreted that such a fast ozone decrease cannot occur due to local chemical processes, but only due to transport.

In the introduction the authors claim that "the mechanisms and environment implications of ozone depletion and gaseous mercury deposition are discussed." A serious discussion of these topics is missing. Is such a discussion at all possible with the presented data?

Minor comments:

Page 1, line 16f: "...the boundary layer ozone and gaseous elemental mercury...": Measurements at the Zeppelin Station do not correspond to the boundary layer.

Page 1, line 27f: "...ozone dropped from typical levels (about 30 ppbv) to few ppbv...": And even below 1 ppbV: see e.g. Helmig et al., JGR 117, D20303, doi: 10.1029/2012JD017531.

Page 2, line 1f: "...considered as possible source of bromide aerosols..."

Page 2, line 4: "...transported over land by monsoon or air turbulence." Already mentioned by referee 1, monsoon and air turbulence are not the correct terms. Why only over land?

Page 2, line 5: "...it is difficult to make detailed chemical observations in source area, ..." I would say it is not more difficult to make these measurements compared to Ny-Alesund, but the access to the source area especially in spring is very limited.

Page 2, line 6: A few more studies in the sea ice area exist. For example, see Jacobi et al., JGE 115, D17302, doi: 10.1029/2010JD013940, 2009; Halfacre et al., ACP 14, 4875-4894, 2014.

Page 2, line 23f: "One of the reasons is that influenced by the North Atlantic Warm Current (NAWC), the near surface air temperature and sea surface temperature (SST) of East Greenland and North of Europe are relatively high...": This statement is unclear.

Page 2, line 29f: "...high level of troposphere BrO can be detected much more frequently in East Arctic (coastal area of north Asia and North America)...": In my opinion this statement is not in agreement with the typical springtime BrO distribution in the Arctic. For example see Theys et al., ACP 11, 1791-1811, 2011.

Page 5, line 10f: I checked the indicated web page for the photographs and was not able to find anything resembling Fig. 10. On the webpage is a folder "Webcam", but the first photos are from 19 May 2015. In the folder "OldWebcam" a photo for 26 April 2015, 22:22:01 (UTC) is available, but with a resolution too low to identify sea ice on the fjord. In the folder "Panorama" photos for the period 22 to 29 April appear to be missing. The authors should verify the source of the used photographs.

Page 5, line 25f: "The occurrence of depleted troposphere ozone and enhanced BrO appears to be unpredictable in May." This statement is unclear.

Page 6, line 17 (and throughout the manuscript): Precipitation of mercury is not the correct term.

Chapter 4.4: In the current form this chapter presents rather limited novelty. I believe it can be deleted or some parts may be merged with previous chapters.

The maps in Fig. 8 were apparently downloaded from the University of Bremen webpage, but the source is not mentioned in the manuscript.

---

## Author Comment (AC1) · 30 Nov 2017

**Response to Anonymous Referee #1**

**Anonymous Referee #1General Comments:**

The paper describes an interesting event of enhanced tropospheric BrO concentrations at Spitsbergen. During that event very rapid decrease of O3 and mercury was observed. In my opinion this is a very interesting case study and should finally be published in ACP. However, in its current form, the paper has a few major and many minor problems. They need to be addressed before acceptance.

**Author's Response:**

We thank the referee for the positive comments for this study and appreciate for all the valuable comments that have improved this manuscript. Please kindly find our point-to-point response to the problems/comments below in blue and the change of the manuscript in orange.

**Major problems**

A) The introduction is not very clear. Please put your study in a better context of existing studies. For example, please make the following points more clear: - what are important current open research questions - what does this study contribute to answer these questions

**Author's Response:**

Thanks for the referee's advices. We have rewritten the introduction part following your suggestions.

[revised manuscript text omitted]

In this study, we have caught a unique process of enhanced bromine and depleted ozone in Ny-Ålesund in late April. The key role of bromine was confirmed by ground-based MAX-DOAS measurements. This event provides a rare opportunity to investigate the source of bromine and process of ozone depletion at this area. Kinetic studies of ozone depletion and gaseous mercury deposition are discussed afterwards."

B) The discussion of the role of long range transport is not clear. In my opinion, from the data you present (trajectory and satellite data), long range transport of air masses enriched in bromine can not be excluded. This should be clearly stated in the paper. Nevertheless, from other findings, I think you can very well conclude that this is probable a local event: -the sea ice occurred only for a very short period close to the

measurements -exactly during this period, the enhanced BrO is observed. Both findings indicate that the enhanced BrO is caused by a local source (connected to the sea ice). Here, it would be very important to have more information: -how large was the area in which the sea ice appeared? -how does this area compare to spatial scale determined by the wind speed (and direction) and the duration of the ozone destruction? -how long have the air masses been in contact with the sea ice before they reached the measurement site? -I think it is in general very important here to discuss the importance of transport compared to chemical processes.

**Author's Response:**

We agree reviewer's opinion that this BrO enhancement is a local event.

Kings Bay is an inlet on the west coast of Spitsbergen, one part of the Svalbard archipelago in the Arctic Ocean. The inlet is 26 km long and 6 to 14 km wide. The BrO map and the BrO VCD in Fig. 6 are both origin from GOME-2 satellite measurements, with a ground-pixel size of $80 \times 40$ km$^2$. According to the light path of the MAXDOAS instrument, the range of MAX-DOAS measurement is about 10km radius area, which covers the central area of the fjord.

The GOME-2 BrO product from University of Bremen showed there is high level of BrO at the North of Greenland as well as Siberia from the mid to late April. The NCEP reanalysis data showed a strong High above Greenland resisted for several days (from 23/04 to 27/04), which indicated the similar conclusion that the transport of the air masses is not within a short period of time. However, what we found by ground-based MAX-DOAS just lasted for several hours, which is at different time scale.

We added some descriptions in the manuscript part 4.1.

"Possible sources of reactive bromine are newly formed sea ice and frost flowers which can provide highly concentrated saline surfaces, thereby adequate sea salt aerosols. Another important source is the transport of the air masses which already contain elevated BrO and depleted ozone. Therefore, we investigate the history of the air masses arriving at measurement site using backward trajectories. Furthermore, the sea ice distribution (Fig.2) and the satellite BrO maps (Fig.10) are important instructions as well.

72-hour backward trajectories at Ny-Ålesund (10, 500, 1000 meters a.s.l.) from 26 April (0600 UTC) to 27 April (1800 UTC) were calculated every 6 hours (Fig.9). From the map of three altitudes, air masses turned from northwest direction, which is origin from North America to the middle of Arctic Sea. From the vertical distribution of air masses, before noon of 26 April, the air masses came from low boundary layer, while after 18:00 26 April, from the upper troposphere.

BrO VCD map from GOME-2 measurements from 20 April to 13 May 2015 are shown in Fig. 10. BrO clouds existed at two main periods: coastal North America and Chukchi Sea during 22-23 April and North of Siberia during 08-11 May 2015. Both of the BrO clouds lasted about three to four days, the first of which was occasionally at the same period with the Ny-Ålesund BrO enhancement event. However, what we

found by ground-based MAX-DOAS just lasted for several hours, which is at different time scale. Thereby, air masses transported from high latitude of Arctic from 22 April might have an impact on BrO enhancement in Ny-Ålesund, but not the most critical reason."

C) You use a fixed Fraunhofer reference spectrum. In my opinion this is not a good strategy here, because you are interested in tropospheric BrO DSCDs. I strongly suggest to use a sequential Fraunhofer reference. The QDOAS software which you use offers this option. Alternatively, you could subtract the BrO DSCDs from the 90◦ measurement of each elevation sequence from the BrO DSCDs at low elevation angles.

**Author's Response:**
   Thanks to the referee's suggestion. The time series of BrO DSCDs of 2 elevation angle are calculated by subtracting the zenith BrO dSCDs. The figure 5 has been revised. The enhancement of troposphere BrO during 26-27 April is still very clear with the highest value of $5.6 \times 10^{14}$ molec/cm$^2$.

[Figure]

D) You apply the method of Sinreich et al. (2013). However, this method should be applied to the tropospheric DSCDs, for which the stratospheric part was already removed (both BrO and O4), see the point above. In the current form, the derived O4 light paths include the light paths in the free troposphere and stratosphere, which are misleading for your study.

**Author's Response:**

Thanks to the referee for pointing this mistake. As shown above, tropospheric BrO has been calculated by subtracting 90 ° dSCDs of the same sequence. However, when calculating $O_4$ DSCD, we find large inconsistency of $O_4$ dSCDs between 2 and 90 elevation angles results, leading to miscalculated results of O4 DSCDs. From the figure below, we can find that the solar zenith angles are from 65 ° to 88 ° at the observation site. As given in Y Wang et al, the rapid method should be applied for SZA<70 ° and RAA>50 ° to ensure that the uncertainties are mainly< 20%. So this method might not very suitable at high latitude polar regions. Therefore, we remove the part of light path calculation and calculate the BrO VMR assuming a homogeneous BrO layer of 1 km thickness at the surface.

[Figure]

E) From the dependence of the BrO DSCDs on elevation angle you conclude that the highest BrO concentration is located at the surface. If this was the case, then the differences of the BrO DSCDs for the low elevation angles should be much larger. In my opinion, the highest BrO concentrations are probably not located directly at the surface. To clarify this important point I suggest to perform one of the following tasks: a) perform RTM simulations (for different assumed BrO profiles) and compare the RTM results to your measurement results. Then you can derive more robust conclusions about the BrO profile shape. b) it would be even better to perform a full profile inversion.

**Author's Response:**

We perform the RTM simulations to get BrO slant columns using four different assumed BrO profiles: a. 0-0.5 km; b. 0-1 km; c. 0-2 km; d. 0.5-1 km; e. 1-2 km. The modeled and measured BrO SCDs during the event were shown in figure below. A

BrO layer height between 0-1 km can be considered as the most possible distribution of BrO layer, which is compatible with the measurements.

The color of the measured BrO dots also showed different steps of this process. The blue dots indicated data points from the first 4 hours of the BrO enhancement event in 26/04, while red and orange dots indicated 4 later hours in the morning of 27/04 when ozone was almost depleted already. The fact that enhanced BrO levels changed from 0-1 km to more likely at 0.5-1 km could be explained by Br/BrO reactions taking place from the boundary layer to the free troposphere.

We modified the corresponding parts of the manuscript.

"We perform the radiative transfer modeling (RTM) simulations using SCIATRAN (Rozanov et al., 2005) to get modeled DAMF using five different assumed BrO profiles with evenly distributed air masses: a. 0-0.5 km; b. 0-1 km; c. 0-2 km; d. 0.5-1 km; e. 1-2 km (Fig.5a). The models are performed under clear sky condition with no aerosol input. Remarkable difference exists for different input profiles. For BrO layer 0-0.5km, 0-1 km and 0-2 km, DAMFs all increase with the decrease of elevation angles. While for BrO layer 0.5-1 km and 1-2 km, the dependence on the telescope elevation angle is weaker especially at small elevation angles.

The modeled BrO SCDs for different input BrO profiles are shown in Fig.5b. The input BrO VCD is $5 \times 10^{13}$ molecules/cm$^2$. The measured BrO SCDs from 20:00 26/04 to 05:00 27/04 are also plotted. The blue dots indicate data points for the first 4 hours in 26/04, while red and orange dots indicate later 4 hours in the morning of 27/04. BrO layer between 0-1 km can be considered as the most possible distribution of BrO layer, which is compatible with the measurements. Thereby, BrO SCDs can be converted to volume mixing ratios (VMR) assuming a homogeneous BrO layer of 1 km thickness at the surface."

[Figure]

Fig. 5 The modeled DAMF (a) and BrO slant columns (b) using radiative transfer modeling simulation. DAMF are the differences of AMF for low elevation angles and zenith direction. The models are performed assuming a clear sky condition with no aerosol. In part b, the tropospheric BrO VCD is $5 \times 10^{13}$ molecules/cm$^2$. The measured BrO SCDs during the event are also shown (solid dots). The blue dots indicated data points from 20:00 to 24:00 in the evening of 26/04. The red and orange dots indicated data points from 00:00-4:00 in the morning of 27/04.

Minor points:

Page 1, line 24: Enhanced BrO was first detected by LP-DOAS observations. Please add the respective references.

**Author's Response:** We add the respective reference.

Page 2, line 4: I think Monsoon and turbulence are not the correct terms here. Monsoon is a tropical to mid latitude phenomenon. Why not simply write 'wind'?

**Author's Response:** It has been revised.

Page 2, line 19: I don't agree with this statement for observations over bright surfaces like ice or snow. I think for such conditions a general underestimation was not reported.

**Author's Response:** This sentence has been revised.

"It is also an important calibration of satellite observations, which has lower spatial and temporal resolution compared with ground-based measurements."

Page 2, line 21: You write 'Long-path DOAS measurement provides regional determination of BrO in PBL.' It is not clear what exactly you want to say with this. LP-DOAS usually has light path lengths between a few and 20 km.

**Author's Response:** This sentence has been removed.

Page 2, line 22: you write 'but few ground-based MAX-DOAS measurement of BrO has been performed in Ny-Ålesund.' I think you can not write this. To my knowledge, University of Bremen performs MAX-DOAS measurements there since many years. (Did you compare your results with the Bremen results?)

**Author's Response:** This sentence has been removed. And corresponding references have been added.

Page 3, line 9: Did you perform MAX-DOAS measurements also before and after the presented period? Have you observed similar events before or after, or in other years?

**Author's Response:** We have carried out the MAX-DOAS measurements at Ny-Ålesund intermittently since 2010. However, the instrument began to work every year mostly from the early summer. So this is the first time we have measured BrO at this area.

Page 3, line 9: What do you mean with 'wavelength adjustment'? Why do you have no measurements during such periods of wavelength adjustment?

**Author's Response:** At that day, we were trying to get some IO information. So we changed another spectrometer with different wavelength range. It is a pity that we do not have continues data records.

Page 3, line 19: Please give typical integration times

**Author's Response:** We have added this information.

"The integration time (typically from 100 ms to 2000 ms multiple 100 scan times) of

each measurement depends on the intensity of scattered light which can be influenced by cloud and visibility."

Page 4, line 17: please add information about typical uncertainties of this method, which are given in Sinreich et al., 2013. You might also have a look at the paper by Wang et al., 2015: Wang, Y., Li, A., Xie, P. H., Wagner, T., Chen, H., Liu, W. Q., and Liu, J. G.: A rapid method to derive horizontal distributions of trace gases and aerosols near the surface using multi-axis differential optical absorption spectroscopy, Atmos. Meas. Tech., 7, 1663-1680, https://doi.org/10.5194/amt-7-1663-2014, 2014.
**Author's Response:** We have given up using this method (see details in major point D).

Page 4, line 20: To my knowledge, the formula is only valid for the tropospheric DSCDs. That means that a) the DSCD for 90◦ elevation of the individual elevation sequence have to be subtracted from the DSCDs of the low elevation angles (for analysis with fixed reference) before the formula is applied b) or the analysis has to be performed with a sequential reference. Since you use QDOAS, I strongly recommend to use this option.
**Author's Response:** As shown above, tropospheric BrO has been calculated by subtracting 90 °dSCDs of the same sequence.

Page 4, line 24: In my opinion it makes no sense to apply such a correction, because of two reasons: a) the differences of the light paths for such small wavelength differences are very small b) the errors of this method (Sinreich et al) are generally rather large. Thus the uncertainties caused by the different wavelengths are negligible. I suggest to remove this part (lines 23 to 31)
**Author's Response:** This part has been removed.

Page 5, line 25: The results in Fig. 5 indicate that enhanced BrO (above the background) is found until 4 May 2015.
**Author's Response:** What the author means is that the surface ozone concentrations did not recover until 29 April. We have revised this in the manuscript.
"At the same period, surface ozone sharply decreased to several ppb and not recovered to normal value until 29 April. During this period, the wind velocity is more than 5 m/s and decreases in 29 April. Over a period of one week, elevated BrO levels went back to the detection limit in 4 May under a stable boundary layer."

Page 5, line 29: I suggest to remove the GOME-2 BrO VCDs from the figure. You show total BrO VCDs. It is unclear how they relate to the BrO mixing ratios from the MAX-DOAS. If you want to keep the GOME-2 BrO VCDs, then you should discuss, how large the stratospheric BrO VCD is. And you should indicate, above which value of the total BrO VCD you think they are affected by enhanced tropospheric BrO.
**Author's Response:** We have removed the GOME-2 BrO VCDs from the Fig.6.

Page 6, line 1: Since you use a fixed Fraunhofer reference, these light path lengths include the light paths in the free troposphere and stratosphere. This makes no sense here. I strongly suggest that you should use a sequential Fraunhofer reference (see major point above).
**Author's Response:** This part has been removed.

Page 6, line 4: at least on 27 April a slight enhancement is seen.
**Author's Response:** The BrO VCD from GOME-2 has been removed from Fig.6.

Page 6, line 9: The differences of the BrO DSCDs between 2∘ and 4∘ are very small (5 to 10%). In my opinion this indicates that the highest values are probably not directly above the surface. I suggest that you should perform RTM simulations (for different assumed BrO profiles) and compare them to the measurement results. Then you can derive more robust conclusions about the BrO profile. Alternatively also a full profile inversion could be done.
**Author's Response:** Thanks for the referee's suggestions. We have performed the RTM simulations and compared them to the measured results. The comparison indicates that BrO is not exponentially decreased with the altitude. Please see detail in major point E and corresponding part in the manuscript.

Page 6, line 12: how high is the boundary layer? Maybe the maximum BrO concentration is on the upper edge of the boundary layer? (see e.g. Wagner et al., 2007)
**Author's Response:** According to the sondes records of temperature, humidity and wind speed from AWIPEV, the height of the troposphere is around 8000 meters and the height of boundary layer is around 1200 meters.

[Figure]

Page 6, line 28: In my opinion this can not be concluded. The trajectories pass over several regions with enhanced BrO (e.g. directly north of Spitsbergen). So, in principle air masses enriched in BrO might have been transported to Spitsbergen.
**Author's Response:** We have revised this part in the manuscript.

"72-hour backward trajectories at Ny-Ålesund (10, 500, 1000 meters a.s.l.) from 26 April (0600 UTC) to 27 April (1800 UTC) were calculated every 6 hours (Fig.9). From the map of three altitudes, air masses turned from northwest direction, which is origin from North America to the middle of Arctic Sea. From the vertical distribution of air masses, before noon of 26 April, the air masses came from low boundary layer, while after 18:00 26 April, from the upper troposphere.

BrO VCD map from GOME-2 measurements from 20 April to 13 May 2015 are shown in Fig. 10. BrO clouds existed at two main periods: coastal North America and Chukchi Sea during 22-23 April and North of Siberia during 08-11 May 2015. Both of the BrO clouds lasted about three to four days, the first of which was occasionally at the same period with the Ny-Ålesund BrO enhancement event. However, what we found by ground-based MAX-DOAS just lasted for several hours, which is at different time scale. Thereby, air masses transported from high latitude of Arctic from 22 April might have an impact on BrO enhancement in Ny-Ålesund, but not the most critical reason."

Page 7, line 15: The enhancement goes very slowly back to normal values (until 4 May). You might mentioned the period here.
**Author's Response:** It has been added in Part 4.1.

Fig. 2b: to which reference (time period) are the anomalies calculated? Please give a reference for the map.
**Author's Response:** Fig. 2b has the same reference as Fig. 2c.

Technical suggestions / language improvement (please note that language improvement is urgently needed; here I list only a few points)
Page 1, line 11: 'difficulty of real-time observations' What exactly is meant here?
**Author's Response:** This sentence has been revised.

Page 1, line 17: separately => respectively?
**Author's Response:** Revised.

Page 2, line 4: former => source?
**Author's Response:** Revised.

Page 2, line 9: If you took this scheme from another publication, please add the corresponding reference.
**Author's Response:** It is an original figure.

Page 4, line 18: The sentence 'The diurnal variations of the clear-sky AMFs for BrO are in accordance with O4 in the boundary layer.' is not clear to me. Please clarify.
**Author's Response:** This part has been removed.

Page 5, line 13: Are you sure that the BrO maps are from NASA? In Fig. 8 you show

results from Uni Bremen.

**Author's Response:** GOME-2 BrO maps are from Uni Bremen. Station overpass BrO VCDs are downloaded from NASA products. We have added the reference.

Page 5, line 26: unpredictable => unexpected?
**Author's Response:** Revised.

Fig. 5: It should be mentioned that presented BrO DSCDs are obtained from measurements at 2° elevation angle.
**Author's Response:** We have added this information in Fig.5.

Fig. 8: Not the tropospheric, but the total BrO VCDs are shown. Please correct the caption. Would it be possible to show additional BrO maps for the period after 27 April?
**Author's Response:** GOME-2 BrO VCDs have been removed from Fig.6.
The BrO maps downloaded from Uni Bremen for period 27/04-13/05 are shown below.

[Figure]

---

## Author Comment (AC2) · 30 Nov 2017

**Response to Anonymous Referee #2**

**Anonymous Referee #2 General Comments:**

Ozone and mercury depletion events are recurrent phenomena characterizing the atmospheric chemistry of both polar regions during springtime. Despite more than 30 years of studies of such events, major gaps still exist in our knowledge on processes, necessary conditions, and amplitude of such events. One of the major difficulties is related to the fact that observed decreases in ozone and mercury concentrations can be caused by local chemical processes as well as the advection of already depleted air masses. Luo et al. report here new measurements of BrO concentrations performed at Ny-Ålesund during a period of approximately three weeks in late April/early May 2015 with an episode of elevated BrO, which is one of the reactive species involved in the chemical destruction of ozone and the oxidation of mercury. Observed ozone and mercury concentrations show strong decreases parallel to the enhancement of BrO. After evaluating meteorological conditions and remote sensing data, the authors propose that this is one of the rare cases with in situ observations of local processes leading to the depletion of ozone and mercury. While such a conclusion appears possible, I'm less convinced than the authors that the available data and analysis allows a firm decision about the influence of local chemistry versus atmospheric transport. I suggest a more extensive discussion of the limitations of the available data and potential scenarios with the corresponding pros and cons, so that each reader can form her/his personal opinion. Below I describe in more detail my major concerns that should be discussed in a revised manuscript.

**Author's Response:**

We sincerely appreciate the referee for the conscientious and responsible comments, which greatly contribute to improving the quality of this manuscript. The source of BrO is the pivotal of the discussion. Following the referee's suggestions, more information and analyses have been included in the manuscript. Please find the point-by-point responses listed below, highlighted in blue and the changes in the manuscript in orange.

**Major comments**

The authors claim that according to the trajectories in Fig. 9 the increase of BrO cannot be explained by the transport since all air masses do have the same origin. Nevertheless, the trajectory close to sea level arriving at the beginning of the BrO episode (18:00; 26/04) travelled closest to the northern coast of Greenland. However, some of the other trajectories shown in Fig. 9 originated from areas close to the North Pole. I believe that this can indicate different origins of the air masses. Moreover, reanalysis data (e.g. NCEP) show that the synoptic situation on that day was characterized by a strong high above Greenland and a large, but weaker low over Siberia. As a result, it appears possible that air masses traveling close to Greenland and influenced by the high may have had different properties and composition compared to those stemming from the North Pole and related to the low. Even

stronger differences are visible in the backward trajectories at 3000 m altitude. (By the way, it remains unclear why these trajectories are shown.) Therefore, the statement that the "trajectories followed similar pathways, which indicate a stable circulation pattern . . ." (page 6, line 27) appears to overly simplifying the meteorological conditions. A more detailed analysis of the mesoscale situation could confirm (or not) the hypothesis of the authors.

**Author's Response:**

In this paper, we have preliminarily analyzed the source of BrO according to the HYSPLIT model and satellite products. Although the temporal resolution of the satellite measurement is one or two overhead data per day, we still can get a clue of the tropospheric BrO distribution. The GOME-2 BrO maps from Uni Bremen showed there is high level of BrO at the North of Greenland as well as Siberia from the mid to late April. However, what we found by ground-based MAX-DOAS just lasted for several hours, which is at different time scale. The NCEP reanalysis data showed a strong High above Greenland resisted for several days (from 23/04 to 27/04), which indicated the similar conclusion that the transport of the air masses is not in a short period of time. Thereby, before, during and after the BrO enhancement episode at the night of 26/04, the air mass of Ny-Ålesund has the similar background level. The back-trajectories showed the air almost come from the same altitude, which means there was no air turbulence at vertical direction during the episode. The back-trajectory at 3000 m has been removed.

According to the referee's suggestions, we add some detail description of the synoptic situation of the observation site in the revised manuscript. We consider that air masses transported from high Arctic might have an impact. But in an hourly time scale, the sea ice occurred in the Kings Bay, exactly the same period the enhanced BrO observed are more likely reason of this event.

Moreover, the BrO map for 26/04 in Fig. 8 demonstrates enhanced concentrations close to Greenland and probably lower concentrations further north (but due to the lack of data this remains speculative). It appears possible that the elevated concentrations are related to the transport of air masses enriched in BrO (or at least air masses enriched in BrO precursors) and already depleted in ozone and mercury. To my knowledge, the most exceptional case of transport-related changes in ozone was presented by Morin et al. (Geophys. Res.Lett., 32, L08809, doi:10.1029/2004GL022098) demonstrating that ozone concentrations can vary on the time scales of minutes due to transport.

**Author's Response:**

The referee thought the BrO map is the indicator of the transport of the air masses. But if you look carefully at the map, you can find that in 27/04 BrO concentration at east side rather than west side of Svalbard archipelago, where Ny-Ålesund located, is at high value. Then, we can take a look at the sea ice distribution on that day. The

west side of Svalbard is ice free ocean, while the east part is still sea ice covered. You can find the same situation at east side of Greenland. Therefore, the local sources from sea ice surfaces are more likely reason of enhanced BrO in this case.

As for the depleted ozone and mercury, there were many earlier studies indicated that they were not always in coincidence with the enhancement of BrO. In some cases, when ozone was only partially depleted and the anticorrelation between BrO and ozone was disappeared, it was possibly caused by the presence of aged air masses already depleted in ozone and mercury (Friess et al, 2011). But in this case, the ozone and mercury concentrations rapidly and synchronously dropped to a very low level, which was more likely caused by a process of reaction in participation with BrO.

The BrO map as well as the BrO VCD further manifests an additional counter-argument of local chemistry driving the ozone and mercury depletion: The Ny-Ålesund area is not part of the area with elevated BrO concentrations. In previous studies claiming local chemical processes (e.g. Jacobi et al. 2006) the measurement sites were located at least close to the area with enhanced BrO. I understand that the authors claim that differences between the satellite and ground measurements of BrO can occur (page 6, line 4ff), but in my opinion such differences, their origin, and potential consequences should be discussed in more detail.

**Author's Response:**

The BrO map and the BrO VCD in Fig. 6 are both origin from GOME-2 satellite measurements, with a ground-pixel size of $80 \times 40$ km$^2$. Besides, the coastal zone is considered as the most difficult area for satellite retrieving because of its complex underlying surface. Therefore, satellite measurements are far from enough to evaluating the local chemical processes of regional area, like Kings Bay area.

Kings Bay is an inlet on the west coast of Spitsbergen, one part of the Svalbard archipelago in the Arctic Ocean. The inlet is 26 km long and 6 to 14 km wide. The effective light path of the ground-based MAX-DOAS is about 10 km (@360nm). The observation direction can cover the central area of the fjord.

Besides, the time resolution of GOME-2 is one or two data per day, while the ground-based MAX-DOAS can provide results every few minutes. Therefore, the ground-based MAX-DOAS measurements are more precise and convincing.

According to the referee #1 suggestion, BrO VCD data in Fig.6 has been deleted in the revised manuscript.

The observations of BrO, ozone, and mercury were not obtained at the same altitude. The authors claim that a large fraction of the enhanced BrO was located close to sea level. First, I agree with referee 1 that the observations in Fig. 7 rather seem to indicate that the highest BrO concentrations were actually at higher altitudes. This needs to be clarified. Second, ozone and mercury were measured at the Zeppelin Station and, thus, at around 480 m a.s.l.. Air masses at Zeppelin Station often represent tropospheric air from higher altitudes and are regularly decoupled from the boundary layer at Ny-Alesund. Thus, the authors need to demonstrate that during the

observed event, such a decoupling between Ny-Alesund and Zeppelin did not occur. Corresponding vertical profiles of meteorological data should be available for example from the French-German AWIPEV Station. Third, the vertical extent of ODEs and elevated BrO can be constrained to only a few hundred meters (e.g. Friess et al., JGR 116, D00R04, doi: 10.1029/2011JD015938). Therefore, the authors need to demonstrate why during this event the observations at sea level and the observations at higher elevations were directly linked. Finally, if the event was local and started at sea level I would expect a delayed response in ozone and mercury at 480 m altitude. However, the time series shown in Fig. 6 suggest either a concomitant increase in BrO and decrease in ozone and mercury or even an onset of the decrease in ozone and mercury before the increase of BrO. This should be clarified. Only if the authors can confirm that increase in BrO and the decrease in ozone occurred in the same air mass, a kinetic analysis as presented in chapter 4.3 is useful (see below).

**Author's Response:**

We perform the RTM simulations to get BrO slant columns using five different assumed BrO profiles with evenly distributed air masses: a. 0-0.5 km; b. 0-1 km; c. 0-2 km; d. 0.5-1 km; e. 1-2 km. The modeled and measured BrO SCDs during the event showed that BrO layer between 0-1 km can be considered as the most possible distribution of BrO layer, which is compatible with the measurements. The color of the measured BrO also showed different steps of this process. The blue dots indicated data points from the first 4 hours of the BrO enhancement event in 26/04, while red and orange dots indicated later 4 hours in the morning of 27/04 when ozone was almost depleted already. The fact that enhanced BrO levels changed from 0-1 km to more likely at 0.5-1 km could be explained by Br/BrO reactions taking place from the boundary layer to the free troposphere where ozone is still present. We modified the corresponding parts of the manuscript.

[Figure]

(The tropospheric BrO VCD is $5 \times 10^{13}$ molecules/cm$^2$)

According to the area of sea ice in Kings Bay and the light path of the MAX-DOAS instrument, the range of this BrO enhancement is about 20 km radius area. The

horizontal distances between measurements location of BrO and ozone/mercury is less than 2 km. The model indicated that BrO is distributed at the layer of 0-1km. The ozone and mercury is measured at 480 m a.s.l.. So it is reasonable to explain the good anticorrelation between BrO dSCD and ozone/mercury.

The elevated BrO can also be several hundred meters (e.g. Wagner et al, ACP-7-3129-2007). As demonstrated in previous part, BrO was distributed at level of 0-1 km. The measurement data themselves are the best prove of the source of this BrO enhancement event.

The meteorology data from AWIPEV Station from 25/04-28/04 2015 are shown:

[Figure]

The authors state that the sea ice shown in Fig. 10 suddenly appeared in the fjord (page 7, line 5f). They further propose that the sea ice is the source of the reactive BrO. This hypothesis seems to be based on previous studies claiming that freshly formed sea ice or first-year sea ice are major sources of reactive bromine. However, the sea ice shown in Fig. 10 does not resemble freshly formed ice. The photograph rather shows crushed pieces of ice stemming from older floes and submerged in sea water. It appears that this sea ice was not formed in the fjord, but rather transported into the fjord by wind and wave actions as mentioned by the authors. Why this type of sea ice should lead to a sustained formation of reactive halogens remains, thus, unclear. I'm also convinced that the claim of the authors that this sea ice type fosters the formation of acidic sea salt aerosols is not warranted (page 7; line 8ff). Although air temperatures are low, the temperature of the crushed ice floes is probably close to the temperature of the water in the fjord, which can only be as low as $\sim$-2$\circ$C. As a result the claimed precipitation of calcium carbonate supporting the acidification of the aerosols formed on the observed sea ice (or even sea water as claimed by the authors; page 7, line 9f) is not likely. In any case, a more detailed description of the ice conditions and how they developed during the days before the event would be necessary and useful.

**Author's Response:**

Low temperature is an important prerequisite for the enhancement of BrO. A correlation between high BrO and ozone depletion with low temperatures was already found in the past researches (e.g. Tarasick and Bottenheim, 2002; Bottenheim et al., 2009; Pöhler et al.,2010; Friess et al, 2011), and was attributed to the temperature

dependence of the thermos-dynamical properties of the ice surfaces, such as the conditions of the quasi-liquid layer and the increase in uptake of HOBr by saline surfaces. Furthermore, model calculations predict that the precipitation of calcium carbonate from sea ice, which occurs 80% below 265 K, is an important prerequisite for the release of BrO since this process facilitates acidification (Sander et al., 2006). So we think the sea ice in the Kings Bay might be the crushed sea ice floating in the bay and transformed to the ice-water mixture in the fjord when it came across sharply dropped temperature.

The maximum Br concentrations derived from the kinetic analysis are higher than the BrO concentrations derived from the DOAS measurements (~45 pptV vs. 15 pptV). First of all, such a kinetic analysis can only be done if it is assured that the measurements concern the same air masses, which is not obvious with the different altitude of the observations (see above). In any case, is this a realistic result? Is the calculated Br too high or the observed BrO too low? This can also be interpreted that such a fast ozone decrease cannot occur due to local chemical processes, but only due to transport.

**Author's Response:** We disagree that. Firstly, the model indicated that BrO is distributed at the layer of 0-1km. The ozone and mercury is measured at 480 m a.s.l.. So it is reasonable to derive the kinetic analysis. Secondly, the kinetics is analyzed assuming the catalytic reactions are dominated by Br+O3→BrO+O2 in a homogenously PBL. It is obviously far from the true situation. Thus, the calculated Br is more likely the sum of all the bromine species, which is larger than the BrO value for sure.

In the introduction the authors claim that "the mechanisms and environment implications of ozone depletion and gaseous mercury deposition are discussed." A serious discussion of these topics is missing. Is such a discussion at all possible with the presented data?

**Author's Response:** We have removed this part.

Minor comments:
Page 1, line 16f: ". . .the boundary layer ozone and gaseous elemental mercury. . .": Measurements at the Zeppelin Station do not correspond to the boundary layer.
**Author's Response:** We think it is within the boundary layer range (~1000 m).

Page 1, line 27f: ". . .ozone dropped from typical levels (about 30 ppbv) to few ppbv. . .": And even below 1 ppbV: see e.g. Helmig et al., JGR 117, D20303, doi: 10.1029/2012JD017531.
**Author's Response:** Revised.

Page 2, line 1f: ". . .considered as possible source of bromide aerosols. . ."
**Author's Response:** Revised.

Page 2, line 4: ". . .transported over land by monsoon or air turbulence." Already mentioned by referee 1, monsoon and air turbulence are not the correct terms. Why only over land?
**Author's Response:** Revised.

Page 2, line 5: ". . .it is difficult to make detailed chemical observations in source area, . . ." I would say it is not more difficult to make these measurements compared to NyAlesund, but the access to the source area especially in spring is very limited.
**Author's Response:** This sentence has been revised.
"As the photochemical reactions are quickly happened and the lifetime of the intermediate products, e.g. the reactive bromine radicals are quite short, more accurate data with higher temporal resolution are needed to analyzing chemical process in PBL and investigating source of bromine."

Page 2, line 6: A few more studies in the sea ice area exist. For example, see Jacobi et al., JGE 115, D17302, doi: 10.1029/2010JD013940, 2009; Halfacre et al., ACP 14, 4875-4894, 2014.
**Author's Response:** Thanks a lot. Correlated references have been added.

Page 2, line 23f: "One of the reasons is that influenced by the North Atlantic Warm Current (NAWC), the near surface air temperature and sea surface temperature (SST) of East Greenland and North of Europe are relatively high. . .": This statement is unclear.
**Author's Response:** The sentence has been revised.
"However, recently the Arctic sea ice coverage has dramatically reduced, especially at East Greenland and North of Europe. Influenced by the North Atlantic Warm Current (NAWC), the near surface air temperature and sea surface temperature (SST) are getting higher at North Europe (Fig.2)."

Page 2, line 29f: ". . .high level of troposphere BrO can be detected much more frequently in East Arctic (coastal area of north Asia and North America). . .": In my opinion this statement is not in agreement with the typical springtime BrO distribution in the Arctic. For example see Theys et al., ACP 11, 1791-1811, 2011.
**Author's Response:** This sentence has been removed.

Page 5, line 10f: I checked the indicated web page for the photographs and was not able to find anything resembling Fig. 10. On the webpage is a folder "Webcam", but the first photos are from 19 May 2015. In the folder "OldWebcam" a photo for 26 April 2015, 22:22:01 (UTC) is available, but with a resolution too low to identify sea ice on the fjord. In the folder "Panorama" photos for the period 22 to 29 April appear to be missing. The authors should verify the source of the used photographs.

**Author's Response:** It is a pity that only OldWebcam can be seen before 19 May 2015.

Page 5, line 25f: "The occurrence of depleted troposphere ozone and enhanced BrO appears to be unpredictable in May." This statement is unclear.
**Author's Response:** This sentence has been revised.
"During 4-5 May, partial ozone (not to near zero level) was depleted in the absence of BrO."

Page 6, line 17 (and throughout the manuscript): Precipitation of mercury is not the correct term.
**Author's Response:** Revised to "deposition".

Chapter 4.4: In the current form this chapter presents rather limited novelty. I believe it can be deleted or some parts may be merged with previous chapters.
**Author's Response:** This part has been merged with previous chapters.

The maps in Fig. 8 were apparently downloaded from the University of Bremen webpage, but the source is not mentioned in the manuscript.
**Author's Response:** Revised.

---

## Referee Report (RR1)

Review of Luo et al "Observations and the source investigation of boundary layer BrO in Ny-Aalesund Arctic"

The paper deals with a case of elevated BrO and depleted O3 which the authors state is driven by local processes. The data are interesting, but several statements are contentious, and the conclusions do not sufficiently take into account alternative interpretations. The paper is worthy of publication in ACP, but these fundamental issues must first be dealt with.

Major/minor concerns:

1) The authors conclusion is that the event described here is a local event, and that the rate of increasing BrO and of decreasing O3 are really fast. Precisely because they are so unusual, it is extremely important to demonstrate beyond doubt that this is a locally-driven event. At the moment the paper does not do this. There are 3 possibilities that I see:

  i)    *That this event is a result of long-range transport.* The event starts at around 17:00 hours on 26 April 2015. This is late in the day, and Global radiation appears to be less than 200W/m2. Trajectory calculations show that the trajectory arriving at 500m agl at noon (the yellow line) travelled at ground level for roughly 2.5 days before rising rapidly to the trajectory end point. The next trajectory had a very different path, reaching the trajectory end point after travelling at roughly 1000m altitude throughout the previous 3 days. i.e. this point in time indicates a discontinuity in air mass origin, and indeed this is reflected in the observational data. The BrO map on 27/4/15 shows an area of enhanced BrO between Svalbard and Greenland, which to me looks as if it is in the path of the trajectory (yellow line) which travelled at the surface. **To help the reader**: Please make the relevant diagrams bigger. I do not see that trajectories arriving at 1000 m asl are relevant, and consider that these could be removed. Also, most of the BrO maps are not needed – the critical ones are 24th to 27th April. Please remove the others and make the ones for 24th to 27th MUCH larger. Then it will be possible to properly compare the trajectories with the BrO maps, and to make sensible assessment of the role of long range transport. If the conclusion is that it's long range transport, we would not need to worry about the low levels of radiation at this time of day.

  ii)   *That this event is driven by the sea ice, but that it is not local to Kings Bay.* The coincidence in timing of sea ice arriving in Kings Bay and in the drop in O3 is very interesting. However, the authors' suggestion seems to be that ozone levels are normal until the ice arrives, and then suddenly it drops. My question here is: why should ozone depletion only "switch on" when the ice arrives in King's Bay..? If the ice is active, wouldn't you expect there to be some sort of equilibrium between the air and the ice, with ozone depletion "travelling with" the sea ice..? Local depletion that has been described before has occurred because the ice has **formed** locally, or air has traversed an area with new sea ice, which is quite different to this case. If the ice is actively depleting ozone, this could have started before the ice arrived in the Bay, in which case air could already be low in ozone and thus be transported, albeit from a local area of depleted ozone. Here the maps of sea ice are critical, and here there is a bit of a problem. The authors provide the web link to the images, but in none of the files can one find the image that is presented in Fig 11. This inconsistency absolutely has to be explained, and the duration of ice in the Bay demonstrated. If a photo is used, the right

and proper citation for the image must be supplied so that readers can look for themselves and assess the local conditions.

iii) *That this event is locally-driven.* However, it must be explained how depletion could occur at the very low solar radiation levels at this time. The authors state on Page 7, line 21, that the heterogenous reactions can still happen under twilight. However, the catalytic cycle shown in Fig 1 is clearly partly photolytic – please provide evidence that there is sufficient light available for the photolytic parts of the catalytic cycle to proceed at a sufficient rate to explain the observed ozone loss.

2) Another major concern is that the location of observations discussed here is not clearly provided. It seems that the MAX-DOAS measuring BrO is located at sea level, and O3 is at 474m on Zeppellin Mountain. Temperature is measured in Ny Alesund. Where is mercury measured? Please state. Also, please show, on Fig 3, the location of Zeppelin Mountain. It seems likely that MAX-DOAS view is over Kings Bay, and that O3, Hg, and temperature are measured in the opposite direction, up on Zeppelin Mountain.

The authors conclude from Fig 5 that the BrO layer 0-1km is the most possible distribution of the BrO. I do not see this so clearly. Please justify this conclusion. Better still, please do the following: Plot Fig 5b as 2 panels, one from 20:00 to 24:00, and one from 00:00 to 04:00 – this will help to clarify where the BrO is, and whether it has moved with time. Also please explain the difference between the red and orange dots? The vertical distribution of BrO is important for the argument of local depletion.

The authors also state (P5 line 27) that wind velocity during this period is more than 5m/s, but a careful look at Fig 6 shows that the wind velocity is highly variable, ranging between ~7m/s and 1 m/s – it is certainly not simply 5 m/s. This must surely affect air mass movement within the Bay… at low wind speeds, one would not expect much vertical mixing, and if a local process is at play, vertical mixing is essential if the signal of depletion (O3) is measured at 474masl…

One minor point, but actually important… If O3 depletion is observed at 474m, there must surely be a lot of sea ice to drive it… I have looked at the web-cam images referred to in the text, and I cannot see evidence of extensive sea ice. Please clarify where the image in Fig 11 came from and provide additional images across this period if possible.

One question, the authors describe that the MAX-DOAS can detect O3… It would help this discussion a lot to show the O3 measured by the MAX-DOAS, even if not very good quality, as it must surely be possible to distinguish between background levels, and none, and this would help with the timing discussion.

3) Fig 7. The authors state on Page 6, line 2, that the differences in BrO dSCD <4 degrees is very small – This is hard to assess as red and orange dots are hard to tell apart. To demonstrate this point, please plot only 2, 3, 4, 5, degrees, and use colours that are easy to distinguish.

4) Was new sea ice actually forming during this event..? And why did the sea ice dissipate.

5) Finally, is it possible to learn anything from the fine structure of Fig 8, e.g. the rise in BrO around 22:00 on 27th April..? This, however, must surely be caused by transport given the lack of solar radiation at this time?

---

## Referee Report (RR2)

Second review of Luo et al "Observations and the source investigation of boundary layer BrO in Ny-Aalesund Arctic"

In my first review, I laid out a number of major concerns. The majority of these have been addressed, and overall the authors have made improvements to their paper. An important aspect to address was the potential role of long-range transport, and the authors have significantly improved the way they address this. They have presented good evidence supporting their view that long-range transport is not responsible for the changes observed in O3 and BrO.

However I still have one major concern that needs to be addressed, which was not adequately tackled in the reviewer's responses. I will re-iterate more clearly below. My other comments are minor points of clarification.

**Major concern:**
A key aspect of this paper is the kinetics calculation of rates of ozone loss. Whether such a calculation is meaningful rests on whether the chemical changes observed are occurring in situ.

In their revised manuscript, the authors continue to argue that the O3 loss and BrO increase are locally-driven, and that for this reason it is possible to calculate kinetic rates of ozone loss.

However, in order to calculate meaningful rates of ozone loss, the chemical processes have to be occurring **in situ**, i.e. **actually happening during the period of observations, and at the place of the observations**. This is where I have a problem with the conclusions as currently presented.

There are various reasons why I am not convinced:
   A) I have looked in as much detail as I could at the new Figure 6 (expanding and stretching it). To me, it looks as if **every single meteorological variable** changes concurrently with the changes in BrO and O3. In particular, wind direction switches from ~350° before the O3 loss/BrO increase, to ~100° during the period of O3 depletion/BrO increase; wind speed increases from ~3 m/s before the O3 loss/BrO increase, to 6 m/s during the period of O3 depletion/BrO increase. After the peak in BrO, both wind speed and wind direction return to their previous speeds/direction. **The fact that changes in all the meteorological variables are concurrent with chemical changes, strongly suggests that changes observed in chemistry are evident because of changes in transport, albeit on a small scale.** The paper would benefit from a figure that showed the range of chemical and meteorological observations – relative humidity, air pressure, temperature, wind velocity, wind direction, ozone, and BrO - from start 26th April to end April 27th to explore in detail what is happening locally. This is the critical period of observations, and none of the current range of figures presents all the information available in sufficient detail.
   B) Key information is presented in section 4.2. The authors state "It is also worth paying attention that the time period that the sea ice existed and the time BrO started to enhance as well as ozone depleted was not exactly the same. From Fig. 8 and 12, the

ozone loss started from 14:00 UTC 26th Apr. And as described upon, the sea ice existed in the fjord after 20:00 UTC 26th Apr." Indeed, Fig 8 shows that BrO enhancement and O3 depletion started at around 14:00, with Fig 12 showing that the sea ice arrived in Kings Bay around 20:00. **Observations of O₃ loss and BrO enhancement thus precede the arrival of ice in the Bay by around 6 hours. By definition, therefore, the observed chemical changes are not happening "in situ", and the observations cannot be used to derive chemical rates of change.**

In particular point B) above leads me to conclude that these data cannot be used to derive O3 loss rates, and that this section of the manuscript should be removed before publication. If the authors wish, they could describe why such a calculation is not feasible. Nonetheless, I believe that the paper is sufficiently interesting to publish without the derivation of O3 loss rates.

**Minor comments:**
i) It is worth saying something for Fig 5, and why the 0-0.5km layer does not best match the data; this fact also points to this not being an in situ process, local to Kings Bay.
ii) Abstract line 1: "presents" should be "presence"
iii) Throughout: "molecular cm$^{-2}$" should be " molec.cm$^{-2}$ "
iv) Abstract line 12: "ice in Kings Bay area, **which** emerged only …"
v) The quality of English needs checking throughout, e.g. "in consistency" is not an English phrase and should be replaced.
vi) Introduction: " A typical heterogenous reaction model between gaseous and condensed phases **is** shown in Fig. 1"
vii) Introduction: "Bromine is released from salty ice surfaces" – but Fig 1 says "aerosol"
viii) Section 2.2 either use dSCD or DSCD but not both.
ix) Towards the end of Section 2.2 "much attention should be paid on the large elevation angles" – define what you mean by "large".
x) Towards the end of Section 2.2: Change "From Fig.5b we can see obviously that the measured BrO DSCDS before midnight are in good consistence with…" to "From Fig.5b we can see obviously that the measured BrO DSCDS before midnight are **best reproduced by**…"
xi) Towards the end of Section 2.2: "**This suggests that the** BrO layer between 0-1km can be considered as the most **likely** distribution."
xii) Section 2.3: "According to the **radio**sonde records of…"
xiii) Section 2.3: "..height of the boundary layer is around 1200 meters at Ny-Alesund" – what is the range of boundary layer height, and is it possible to say what was it on 26th April at the start of the O3 loss/BrO increase?
xiv) Section 2.3: The trajectories shown are not "ensemble" trajectories – remove the word "ensemble" at the end of section 2.3.
xv) Section 4.1, first paragraph "Then we calculated the air mass backwards trajectory ending at 18:00 (UTC) 26th April in every hour (Fig 9**b**). i.e. not Fig 9a here.
xvi) Section 4.2 – "The ice-sea water mixture was filled in the gaps, which was salty-enriched." – What evidence do you have that it was salty..??

xvii)    Section 4.2, second paragraph – if more than 80% of carbonate precipitates, will it make things acid, or only less alkali..?? Why should they become acid?

xviii)    Section 4.2, second paragraph – "This process will provide acid aerosol from sea water" – do the authors really mean it will produce *aerosol*..?? If so, what is the mechanism..? What evidence do the authors have that the surface is airborne..? Throughout the majority of the paper they refer to sea-ice… Which surface is the one that matters..?

xix)    Section 4.2 – If the authors still want to discuss influence of temperature with respect to ozone loss, they should refer to previous work looking at the link between these processes, e.g. Tarasik and Bottenheim, ACP 12, 197, 2002. Note that Tarasik and Bottenheim suggest -20C is the temperature able to trigger ODEs.

xx)    Section 4.2 – again, the authors write "The sea ice is not totally fresh ice but the low air and water temperature in the fjord might cause the formation of brine ice mixture which is rich in sea salt aerosols" – the brine may be rich in sea salt, but aerosol only refers to sea salt once airborne – please correct this.

xxi)    Conclusions – again, the authors refer to "low temperature provide acid aerosols" – do they really mean that the surfaces are airborne? This needs clarification or correction

xxii)    Conclusions – statements about kinetics calculations need to be removed, as per Major concern described above.

xxiii)    Fig 2 b and c – maps are poor quality and need to be improved.

xxiv)    Fig 5 – caption – The modelled DAMF (a) and BrO slant columns (b) – but (b) is now DSCD … Also, is fig c) SCD or DSCD?

xxv)    Fig 10 – quality is somewhat improved with the large images, but they are still hard to read. Please improve, and indicate location of Spitzbergen.

xxvi)    Fig 11 – needs information on source of photo, in particular to clarify that it is not the Kings Bay web cam.

---

## Editor Decision (ED1)

**Editor's comments:**

Reviewer 3 suggested changes to the manuscript, that were needed to deal with the question whether or not the data can be used to demonstrate that ozone loss and bromine enhancements were "very fast". The authors initially argued that the changes were happening *in situ,* caused by local chemical processes, and were thus surprisingly fast. However, Reviewer 3 (supported by earlier comments from Reviewer 1) concluded that the reason the changes were fast was because they were driven by transport, something which has been reported previously in the literature. The authors made some requested changes. However, they still present their results on the basis that the rate of change is surprising – if it is transport-driven, the rates of change are not surprising, and should not be over-stated; indeed such changes and rates of change, have been reported in previous papers. Therefore amendments to the manuscript are still needed in a number of places, to adjust the tone of their paper to reflect this fact.

There are two ways to address the major concerns raised. One would be the following:

i) Remove the sentence (**page 1 line 18 to 20**) that states: "the ozone loss rate during the bromine enhancement period was 10.3 ppbv h$^{-1}$, which is extremely high compared to those observed in other areas"

ii) On **page 2, line 32**, please change the statement " a unique process event" and replace with "an event"

iii) **Page 8 line 4 and 5**, change the sentence: "The concurrent changes in the chemical and meteorological variables demonstrate the impact of environment change on this ozone depletion/BrO event" and replace with "The concurrent changes in the chemical and meteorological variables demonstrate that changes in observed chemistry are evident because of changes in transport, albeit on a small scale".

iv) **Page 8 line 17 to 19**, please remove the first three sentences of this section; the section should therefore start "The deposition of gaseous mercury…."

v) **Page 8 line 23**, remove the statement "The mercury loss rate is ~25 ngm$^{-3}$h$^{-1}$ or 6 ngm$^{-3}$d$^{-1}$

vi) In the conclusions section, **Page 9 lines 2 to 7,** please remove all the text from "The concurrent changes in chemical and meteorological variables…….Further observations are required to identify its chemical mechanisms"

vii) **Page 9 line2**, alter the text to read: "By analysing the air mass history and sea ice conditions, this BrO enhancement event was found to more likely be a regional process, driven by changes in sea ice and transport on a local scale."

viii) **Table 2** should also be removed.

The alternative is that the authors ensure that every time they refer to the rate of changes calculated, they clearly state that these are most likely driven by transport, and then compare the calculated rates of change to those observed by others, and published in the literature. The context is extremely important.

I note also the Author's reply to the minor comment vii), regarding aerosol as surfaces for heterogenous reactions. The authors are correct, that sea salt aerosol has been shown to be an important source of bromine compounds to the atmosphere. However, they confuse the range of different surfaces which can act as a source. For example, in general, salty condensed phases can be a source of bromine compounds, and this can be achieved directly, and not only via producing

aerosol. For example, there is evidence now that frost flowers are not a source of aerosol, however they could still be a modest source of bromine compounds directly to the atmosphere. To address this the authors should remove mention of aerosol on **page 2 lines 6 and 8**. On **page 6 line 25**, they should change wording to read: "…frost flowers, which can provide highly concentrated saline surfaces, and also sea salt aerosol." They should also remove the label "aerosol" in **Fig 1**. My view is that the discussion of bromine sources would then be consistent.

---

## Author Response (AR2)

**Response to Anonymous Referee #1**

Thanks a lot for the referee #1's comments. Please kindly find the author's responses (in blue).

**Anonymous Referee #1Comments:**

The paper is much improved, and all my major points were addressed.

However, for two of the major points still some corrections are suggested, see details below. After these points are addressed, I recommend this paper for publication.

1) Previous major point B:

I still think the possibility of long range transport cannot be completely ruled out (but I agree with the authors that it very probably plays no important role for the observed event). I suggest to change the logic of the discussion about the role of long range transport. In my view the following points are important:

a) In 2015 Spitsbergen is close to the sea ice edge; only in the South of Spitsbergen the sea is free of ice. This means that the sensitivity of satellite observations for the detection of enhanced BrO is high north of Spitsbergen.

b) In the GOME BrO maps enhanced BrO VCDs are observed north and east of Spitsbergen during the period of interest and the days before. However, directly north of Spitsbergen always an area with low BrO VCDs is observed from satellite.

c) Trajectory calculations show that transport from the north takes place. However, it is not very probable that the enhanced BrO over the Kings bay is caused by this long range transport because of the rather low BrO VCDs directly north of Spitsbergen. Also such long range transport would probably have led to a longer period of enhanced BrO.

d) Trajectory calculations show that transport from the east coast of Greenland is not probable. So transport from these areas of enhanced BrO VCDs can very porbably be ruled out.

e) In summary, long range transport as an explanation for the enhanced BrO at Spitsbergen can not be completely ruled out. But given the temporal coincidence of the enhamced BrO and the occurrence of the sea ice at Spitsbergen, it can be concluded that the local production of BrO is the most probable explanation.

In addition I have two suggestions:

-please add trajectory calculations also for 2 days before and 2 days after the event. In this way long range transport patterns could be compared with those on the day of the event.

-In the text you write: 'BrO VCD maps from GOME-2 measurement from 20 April to 13 May are shown in Fig. 10.' But in Fig. 10 measurements start only on April 24. Please make the text and Fig. consistent.

**Author's Response:**

   Thanks for the referee's suggestion. We have improved the discussion part. Please kindly find the change in the revised manuscript part 4.1.

   Considering the resolution of the figures, the trajectory calculations for 2 days before and 2 days after the event, as well as the BrO VCD maps from GOME-2 are shown in the appendix part. Only maps from $25^{th}$ to $28^{th}$ are shown in the manuscript.

2) Previous major point E:

In Fig. 5 you show DAMFs in the left (top) figure, but SCDs in the right (bottom) figure. Please make both figures consistent. My suggestion would be to simply subtract the 90 ° values from the low elevation values in the right (bottom) figure (then the 90 ° values of the simulated and measured data should both be zero). You might also use the same x-axes for both figures.

**Author's Response:** Fig. 5 has been revised following referee's suggestions. From the figure, it is more convinced that distribution of BrO is in accordance with model of layer 0-1 km during the enhancement.

[Figure]

Fig.5 The modeled DAMF (5a) and BrO DSCD (5b) using radiative transfer modeling simulation and the measured BrO DSCDs (5c) from 26/04 20:00 to 27/04 04:00. DAMF are the differences of AMF for low elevation angles and zenith direction. The models are performed assuming a clear sky condition with no aerosol. In part b, the tropospheric BrO VCD is $5 \times 10^{13}$ molecules/cm$^2$. The color codes of the measured BrO DSCDs which are also shown in 5b (solid dots) are put into one-to-one correspondence to dots in 5c.

**Response to Anonymous Referee #2**

It is a pity that the referee #2 has refused the manuscript. We are so sorry that color codes of BrO SCD in Fig.5 might have led to a lot of misunderstandings. Anyway, thanks a lot for the referee #2's comments. And please kindly find the author's responses (in blue).

**Anonymous Referee #2 Comments:**

The authors put effort into a better interpretation of the MAX-DOAS data concerning the horizontal distribution of BrO. They added the new figure 5 including the results for some radiation transfer calculations and a comparison between observed BrO slant columns as a function of the viewing angle of the instrument and simulated slant columns for different layers of enhanced BRO also as a function of the viewing angle. This corresponds to the additional information requested by both reviewers. Unfortunately, the presentation of these results and the derived conclusions are rather confusing and in my opinion counteract the conclusion of a local ozone depletion event:

1). The authors present the observed BrO slant columns with at least three different color codes, but refer in the title two only two categories (4 hours before or after midnight). Why is that? Why not using only two colors?

**Author's Response:** We have revised the Fig.5 and added some explanations. The 90 $^{\circ}$ values of BrO SCDs are subtracted so that the simulated and measured BrO DSCDs of 90 are all zero. Then we find an even better correlation between modeled BrO DSCDs for layer 0-1km and the measured ones during the enhancement. The color codes refer to the time series of BrO DSCDs from about April 26[th] 20:00 to April 27[th] 4:00. The temporal resolution is about two minutes.

[Figure]

Fig.5 The modeled DAMF (5a) and BrO DSCD (5b) using radiative transfer modeling simulation and the measured BrO DSCDs (5c) from 26/04 20:00 to 27/04 04:00. DAMF are the differences of AMF for low elevation angles and zenith direction. The models are performed assuming a clear sky condition with no aerosol. In part b, the tropospheric BrO VCD is $5 \times 10^{13}$ molecules/cm$^2$. The color codes of the measured BrO DSCDs which are also shown in 5b (solid dots) are put into one-to-one correspondence to dots in 5c.

2). What is the reason for using 4-hr averages, although according to figure 13 the entire depletion happened within less than 6 hours with potentially highly variable Br and BrO concentrations?

**Author's Response:** Highly variable BrO concentrations are shown with temporal resolution of several minutes. No average of BrO DSCD is calculated. Please see Fig. 5 in previous response.

3). With the current figure it is difficult to identify, however, it seems to me that the observations before and after midnight are rather different. Why do the authors then claim that one profile with BrO in the 0-1 km range can explain all observations? Why don't the authors use the highest temporal resolution possible for the observations, which is probably below 30 min for a full scan of all viewing angles?

**Author's Response:** It is exactly a full scan of all viewing angles with highest temporal resolution of about 2 minutes. From the Fig. 5c, the time series of BrO DSCDs are one-to-one correspondence with Fig. 5b. It is more convinced that distribution of BrO is in accordance with model of layer 0-1 km during the enhancement. Indeed, the 0-1 km range is still a rough estimation of BrO distribution. But it helps to understand the relationship between ozone loss and BrO enhancement in this event.

4). Why do the authors only show results for this specific period? It would be interesting to see the results also for low BrO slant columns.

**Author's Response:** From Fig.6 and 7, we can find that only within the enhancement event in 26-27 April BrO DSCDs in each elevation angle are differed. While other data is at low level and cannot distinguish between each elevation angle. So it is of less meaning to show the low BrO DSCDs.

5). The authors claim that the observations shown in figure 5 are best reproduced by the BrO profile with a homogeneous layer between 0 and 1 km altitude. Why is that? Fig. 5b actually shows only two simulations with slant columns at $2°$ outside the observed range: BrO in a layer from 0 to 0.5 km and BrO in a layer from 0 to 1 km. The authors do not provide any quantitative information regarding the comparison of the simulations and the results. I find that the best agreement for the period before midnight (blue points) is obtained with the profile with BrO in the layer from 0 to 2 km and for the period after midnight with the profile with BrO in the layer from 0.5 to 1 km. However, this is only based on the visual inspection of the figure. In my opinion, the comparison does not support the claim of the authors that the observed event corresponds to a local BrO formation caused by processes at the surface since

the simulated slant columns for a BrO layer from 0 to 0.5 km are far from all observations at low elevation angles.

**Author's Response:** The BrO DSCDs are used instead of BrO SCDs in Fig.5 in order to better understand the distribution of BrO by comparing the modelled and measured BrO columns. Please find the new Fig. 5 in previous response. It has to be noted that the inaccuracy of modelled BrO is getting larger at lower elevation angles. So much attention should be paid on the large elevation angles. From Fig.5b, we can see obviously the measured BrO DSCDs before midnight are in good consistence with model for layer 0-1 km.

In summary, the analysis of the presented simulations and observations are to superficial to support the claim of the authors that (1) the simulation with BrO confined to 0 to 1 km corresponds best to the observations and (2) that the BrO increase is a local event. In my opinion, many other scenarios seem equally possible or correspond even better to the presented data.

Regarding the other points raised in my previous report (trajectory analysis, analysis of the mesoscale situation, transport of BrO from the Arctic Ocean, analysis of the boundary layer and impact on the observations, timing of the increase in BrO and decrease of ozone and mercury, sea ice conditions, impact of temperature, significance of calcium carbonate precipitation, measured BrO versus estimated Br) the revised manuscript does not provide a lot of additional information. The authors provide some arguments in the "Author's Response", but most of them are not included in the revised manuscript. Overall, the tone of the manuscript is still the same of the original version, since the authors claim throughout the manuscript that the depletion was a local process without providing a balanced analysis of other possibilities.

**Author's Response:**
Please find the details in the revised manuscript discussion part.

**Response to Anonymous Referee #3**

**Anonymous Referee #3 General Comments:**

Review of Luo et al "Observations and the source investigation of boundary layer BrO in Ny-Aalesund Arctic"

The paper deals with a case of elevated BrO and depleted O3 which the authors state is driven by local processes. The data are interesting, but several statements are contentious, and the conclusions do not sufficiently take into account alternative interpretations. The paper is worthy of publication in ACP, but these fundamental issues must first be dealt with.

**Author's Response:**

We thank the referee for the positive comments for this study and appreciate for all the valuable comments that have improved this manuscript. Please kindly find the author's responses (in blue).

Major/minor concerns:

1) The authors conclusion is that the event described here is a local event, and that the rate of increasing BrO and of decreasing O3 are really fast. Precisely because they are so unusual, it is extremely important to demonstrate beyond doubt that this is a locally-driven event. At the moment the paper does not do this. There are 3 possibilities that I see:

i) That this event is a result of long-range transport. The event starts at around 17:00 hours on 26 April 2015. This is late in the day, and Global radiation appears to be less than 200W/m2. Trajectory calculations show that the trajectory arriving at 500m agl at noon (the yellow line) travelled at ground level for roughly 2.5 days before rising rapidly to the trajectory end point. The next trajectory had a very different path, reaching the trajectory end point after travelling at roughly 1000m altitude throughout the previous 3 days. i.e. this point in time indicates a discontinuity in air mass origin, and indeed this is reflected in the observational data. The BrO map on 27/4/15 shows an area of enhanced BrO between Svalbard and Greenland, which to me looks as if it is in the path of the trajectory (yellow line) which travelled at the surface. **To help the reader:** Please make the relevant diagrams bigger. I do not see that trajectories arriving at 1000 m asl are relevant, and consider that these could be removed. Also, most of the BrO maps are not needed – the critical ones are 24th to 27th April. Please remove the others and make the ones for 24th to 27th MUCH larger. Then it will be possible to properly compare the trajectories with the BrO maps, and to make sensible assessment of the role of long range transport. If the conclusion is that it's long range transport, we would not need to worry about the low levels of radiation at this time of day.

ii) That this event is driven by the sea ice, but that it is not local to Kings Bay. The coincidence in timing of sea ice arriving in Kings Bay and in the drop in O3 is very interesting. However, the authors' suggestion seems to be that ozone levels are normal until the ice arrives, and then suddenly it drops. My question here is: why should ozone depletion only "switch on" when the

ice arrives in King's Bay..? If the ice is active, wouldn't you expect there to be some sort of equilibrium between the air and the ice, with ozone depletion "travelling with" the sea ice..? Local depletion that has been described before has occurred because the ice has formed locally, or air has traversed an area with new sea ice, which is quite different to this case. If the ice is actively depleting ozone, this could have started before the ice arrived in the Bay, in which case air could already be low in ozone and thus be transported, albeit from a local area of depleted ozone. Here the maps of sea ice are critical, and here there is a bit of a problem. The authors provide the web link to the images, but in none of the files can one find the image that is presented in Fig 11. This inconsistency absolutely has to be explained, and the duration of ice in the Bay demonstrated. If a photo is used, the right and proper citation for the image must be supplied so that readers can look for themselves and assess the local conditions.

iii) That this event is locally-driven.   However, it must be explained how depletion could occur at the very low solar radiation levels at this time. The authors state on Page 7, line 21, that the heterogenous reactions can still happen under twilight. However, the catalytic cycle shown in Fig 1 is clearly partly photolytic – please provide evidence that there is sufficient light available for the photolytic parts of the catalytic cycle to proceed at a sufficient rate to explain the observed ozone loss.

**Author's Response:**

The reviewer discussed three possibilities of this event clearly and deeply in the comment. We appreciate reviewer's hard work and accept many helpful suggestions to improve the manuscript.

We analyze from the following three aspects:

**1)  The air mass origin**

In order to find the detail of the air mass origin, we also calculate the air mass back trajectory in 26th April using HYSPLIT model. It shows air mass at 500 m altitude has different origin before/after 15:00 UTC 26 Apr. The wind direction changed to north with higher velocity. After then, the air mass has a relatively stable origin from 1000 m height.

From the GOME-2 BrO VCD maps, we can find enhanced BrO are observed at east of Greenland (Red box), far north of Siberia (Blue circle) and east of Spitsbergen (Black box) during the period of interest and the days before.

Combining the above information, firstly, trajectory calculations show that transport from the east coast of Greenland and east coast of Spitsbergen are not possible. So transport from these areas of enhanced BrO can be ruled out. Secondly, trajectories also show that after 16:00 UTC 26 Apr transport from the north takes place, which means the high BrO in the blue circle might have influenced this event. However, we have to notice that a). the altitude of air mass is up to 1000 meters; b). there is no enhancement along the path; c). the time scale is unreasonable. The BrO enhancement we found by ground-based MAX-DOAS as well as ozone loss just

lasted for several hours. But the high level of BrO in the blue circle area lasted more than one day. If the high BrO air mass transported from blue circle area, why there is no enhancement along the path?

[Figure]

**2) The sea ice origin**

Firstly, the MODIS remote sensing product and zeppelin webcam have verified

that west of Spitsbergen as well as Kongsfjorden were sea ice free area during April 2015.

Secondly, the sea ice image presented in Fig. 11 is taken by author at 21:00 UTC 26th Apr, which is an unusual phenomenon in the fjord. It is a pity that the zeppelin webcam did not have a clear record of this sea ice process. So we estimated that the sea ice is formed or floated in the fjord after 20:00 UTC 26th Apr.

Now the point is if the sea ice is newly formed or just floated from other area. From the shape of ice in Fig. 11, the sea ice is not looked like newly formed sea ice because of crashed pieces and corrugated edge. So we consider that the sea ice was formed before floating in the bay and transformed to the ice-water mixture when it came across sharply dropped temperature.

3) **The ozone depletion**

We have changed the time scale to a standard UTC time in each figure, which is very important to analyse this special case of ozone depletion process. In this case, ozone depletion has the anti-correlation with BrO and the ozone loss rate is extremely high compared with previous researches. Taking the ozone loss in Apr 21st, 2015 as an comparison (Figure below), the ozone loss process continued for three days and the ozone loss rate is about 2.5 ug $m^{-3}h^{-1}$. The wind velocity is so high (>10 m/s), and the wind direction is almost unchanged. The air temperature also dropped to very low, which has a good correlation with the ozone concentration. All the evidence showed that the process in Apr 21st is a long-range transport process. But the case in Apr 26th is quite different. The ozone loss rate is much faster while the whole period is quite short instead. The wind velocity is highly variable between 1-7 m/s with unstable wind directions and mixing height.

It is also worth paying attention that the time period that the sea ice existed and the time BrO started to enhance as well as ozone depleted was not exactly the same. From Fig.8 and 12 in the manuscript, the ozone loss started from 14:00 UTC 26th Apr. And as described upon, the sea ice existed in the fjord after 20:00 UTC 26th Apr. It makes the synchronizing variation of BrO and ozone as well as the distribution of 0-1 km reasonable.

[Figure]

In conclusion, we think the sea ice rather than the long-range transport is the main reason of this event. The sea ice is not totally fresh ice but the low air and water temperature in the fjord might cause the formation of brine ice mixture, which is rich in sea salt aerosols. The sea ice in the fjord is not the trigger of the ozone loss because the ozone loss is occurred earlier than the existence of sea ice.

2) Another major concern is that the location of observations discussed here is not clearly provided. It seems that the MAX-DOAS measuring BrO is located at sea level, and O3 is at 474m on Zeppellin Mountain. Temperature is measured in Ny Alesund. Where is mercury measured? Please state. Also, please show, on Fig 3, the location of Zeppelin Mountain. It seems likely that MAX-DOAS view is over Kings Bay, and that O3, Hg, and temperature are measured in the opposite direction, up on Zeppelin Mountain.

**Author's Response:**

The locations of the observations are described in part 2.3. MAX-DOAS can measure target trace gases in the troposphere. The analysis of BrO distribution concludes that BrO is located at 0-1km in the troposphere. The BrO results represented the average value of 0-1 km. Ozone and mercury is measured on Zeppelin Mountain, which is at altitude of 480 m a.s.l., representing for the background level of this area. Temperature is from AWI records, which is at ground level. The Zeppelin Station has been marked in Fig.3.

The authors conclude from Fig 5 that the BrO layer 0-1km is the most possible distribution of the BrO. I do not see this so clearly. Please justify this conclusion. Better still, please do the following: Plot Fig 5b as 2 panels, one from 20:00 to 24:00, and one from 00:00 to 04:00 – this will help to clarify where the BrO is, and whether it has moved with time. Also please explain the difference between the red and orange dots? The vertical distribution of BrO is important for the argument of local depletion.

**Author's Response:**

Fig. 5 has been revised. The 90 ° values of BrO SCDs are subtracted so that the simulated and measured BrO DSCDs of 90 ° are all zero. Then we find an even better correlation between modeled BrO DSCDs for layer 0-1km and the measured ones during the enhancement. The color codes refer to the time series of BrO DSCDs from about April 26th 20:00 to April 27th 4:00. The temporal resolution is about two minutes. It has to be noted that the inaccuracy of modelled BrO is getting larger at lower elevation angles. So much attention should be paid on the large elevation angles. From Fig.5b, we can see obviously the measured BrO DSCDs before midnight are in good consistence with model for layer 0-1 km.

[Figure]

Fig.5 The modeled DAMF (5a) and BrO DSCD (5b) using radiative transfer modeling simulation and the measured BrO DSCDs (5c) from 26/04 20:00 to 27/04 04:00.
DAMF are the differences of AMF for low elevation angles and zenith direction. The models are performed assuming a clear sky condition with no aerosol. In part b, the tropospheric BrO VCD is $5 \times 10^{13}$ molecules/cm$^2$. The color codes of the measured BrO DSCDs which are also shown in 5b (solid dots) are put into one-to-one correspondence to dots in 5c.

The authors also state (P5 line 27) that wind velocity during this period is more than 5m/s, but a careful look at Fig 6 shows that the wind velocity is highly variable, ranging between ~7m/s and 1 m/s – it is certainly not simply 5 m/s. This must surely affect air mass movement within the Bay… at low wind speeds, one would not expect much vertical mixing, and if a local process is at play, vertical mixing is essential if the signal of depletion (O3) is measured at 474masl…

**Author's Response:**

We agree the reviewer's point of view. Especially when compared to the Apr 21$^{st}$ process, which is a typical long-range transport process, this is a regional event occurred at this area.

One minor point, but actually important… If O3 depletion is observed at 474m, there must surely be a lot of sea ice to drive it… I have looked at the web-cam images referred to in the text, and I cannot see evidence of extensive sea ice. Please clarify where the image in Fig 11 came from and provide additional images across this period if possible.

**Author's Response:**

Please see author's previous response in "2) The sea ice origin". As we know, the area of Kings Bay is about 26 km long and 6 to 14 km wide. At that day, the sea ice floated into the bay and constantly covered the area of the bay. But since no clear webcam evidences, it can just be a reference.

One question, the authors describe that the MAX-DOAS can detect O3… It would help this discussion a lot to show the O3 measured by the MAX-DOAS, even if not very good quality, as it must surely be possible to distinguish between background levels, and none, and this would help with the timing discussion.

**Author's Response:**

The MAX-DOAS can detect the total column of ozone. But since the stratospheric ozone is far more than tropospheric ozone, it is usually used to calculate stratospheric ozone (the altitude of "ozone layer"). It is difficult to identify the variations of tropospheric ozone by ground-based MAX-DOAS independently.

3) Fig 7. The authors state on Page 6, line 2, that the differences in BrO dSCD <4 degrees is very small – This is hard to assess as red and orange dots are hard to tell apart. To demonstrate this point, please plot only 2, 3, 4, 5, degrees, and use colours that are easy to distinguish.

**Author's Response:**

We have revised Fig.7. The BrO DSCD of 2, 3, 4 degrees are shown in the upright plot.

[Figure]

4) Was new sea ice actually forming during this event..? And why did the sea ice dissipate.

**Author's Response:**

From the shape of ice, it is not looked like newly formed sea ice because of crashed pieces and corrugated edge. But the formation of the ice-water mixture is reasonable because of the low temperature. The saline-ice mixture is also one of the sources of the BrO.

5) Finally, is it possible to learn anything from the fine structure of Fig 8, e.g. the rise in BrO around 22:00 on 27th April..? This, however, must surely be caused by transport given the lack of solar radiation at this time?

**Author's Response:**

The amount of ozone can be the restriction of the BrO reaction. During the BrO enhancement event, BrO and ozone were maintaining equilibrium between each other. When ozone dropped to the lower limit of the reaction, the reaction of $Br+O_3\rightarrow BrO+O_2$ would stop (the situation at 26th night). When ozone recovered to a certain level, the reaction restarted. And if there is still enough bromine, ozone will drop again until the reaction is finished. Therefore, the rise of BrO at 22:00 on 27th Apr might be another proves that there is enough sea ice to generate reactive bromine until it melted.

[revised manuscript text omitted]

---

## Author Response (AR3)

**Second Response to Anonymous Referee #3**

Second review of Luo et al "Observations and the source investigation of boundary layer BrO in Ny-Aalesund Arctic":

In my first review, I laid out a number of major concerns. The majority of these have been addressed, and overall the authors have made improvements to their paper. An important aspect to address was the potential role of long-range transport, and the authors have significantly improved the way they address this. They have presented good evidence supporting their view that long-range transport is not responsible for the changes observed in O3 and BrO.

However I still have one major concern that needs to be addressed, which was not adequately tackled in the reviewer's responses. I will re-iterate more clearly below. My other comments are minor points of clarification.

**Author's Response:**

We are very grateful for the referee's valuable comments that have improved this manuscript. Please kindly find the author's responses below.

**Major concern:**

A key aspect of this paper is the kinetics calculation of rates of ozone loss. Whether such a calculation is meaningful rests on whether the chemical changes observed are occurring in situ.

In their revised manuscript, the authors continue to argue that the O3 loss and BrO increase are locally-driven, and that for this reason it is possible to calculate kinetic rates of ozone loss.

However, in order to calculate meaningful rates of ozone loss, the chemical processes have to be occurring **in situ**, i.e. **actually happening during the period of observations, and at the place of the observations**. This is where I have a problem with the conclusions as currently presented. There are various reasons why I am not convinced:

A) I have looked in as much detail as I could at the new Figure 6 (expanding and stretching it). To me, it looks as if **every single meteorological variable** changes concurrently with the changes in BrO and O3. In particular, wind direction switches from ~350$^o$ before the O3 loss/BrO increase, to ~100$^o$ during the period of O3 depletion/BrO increase; wind speed increases from ~3 m/s before the O3 loss/BrO increase, to 6 m/s during the period of O3 depletion/BrO increase. After the peak in BrO, both wind speed and wind direction return to their previous speeds/direction. **The fact that changes in all the meteorological variables are concurrent with chemical changes, strongly suggests that changes observed in chemistry are evident because of changes in transport, albeit on a small scale.** The paper would benefit from a figure that showed the range of chemical and meteorological observations – relative humidity, air pressure, temperature, wind velocity, wind direction, ozone, and BrO - from start 26th April to end April 27th to explore in detail what is happening locally. This is the critical period of observations, and none of the current range of figures presents all the information available in sufficient detail.

**Author's Response:**

The chemical and meteorological information from the start of 26$^{th}$ April to noon on 28$^{th}$ April are shown together in Figure 12. When the ozone depletion/BrO enhancement occurs, the air temperature continuously decreases, and the relative humidity drops from 80% to less than 65%,

with the wind direction switching from northwest to east. The concurrent changes in the chemical and meteorological variables demonstrate the impact of environment change on this ozone depletion/BrO enhancement event.

[Figure]

Fig. 12 Time series of the chemical and meteorological changes during the BrO enhancement event, blue triangle presents the sea ice existence period in Kings Bay

B) Key information is presented in section 4.2. The authors state "It is also worth paying attention that the time period that the sea ice existed and the time BrO started to enhance as well as ozone depleted was not exactly the same. From Fig. 8 and 12, the ozone loss started from 14:00 UTC 26th Apr. And as described upon, the sea ice existed in the fjord after 20:00 UTC 26th Apr." Indeed, Fig 8 shows that BrO enhancement and O3 depletion started at around 14:00, with Fig 12 showing that the sea ice arrived in Kings Bay around 20:00. **Observations of O3 loss and BrO enhancement thus precede the arrival of ice in the Bay by around 6 hours. By definition, therefore, the observed chemical changes are not happening "in situ", and the observations cannot be used to derive chemical rates of change.**

In particular point B) above leads me to conclude that these data cannot be used to derive O3 loss rates, and that this section of the manuscript should be removed before publication. If the authors wish, they could describe why such a calculation is not feasible. Nonetheless, I believe that the paper is sufficiently interesting to publish without the derivation of O3 loss rates.

**Author's Response:**

Considering the referee's suggestions, we have removed the calculation part. Discussions on the ozone and GEM depletion rate are reserved.

**Minor comments:**

i) It is worth saying something for Fig 5, and why the 0-0.5km layer does not best match the data; this fact also points to this not being an in situ process, local to Kings Bay.

**Author's Response:** We have added some explanations in part 3.

"The measured BrO DSCDs best match the model for the 0-1 km layer during the enhancement, which means that the BrO enhancement event was a regional rather than an in situ process."

ii) Abstract line 1: "presents" should be "presence"
**Author's Response:** Done.

iii) Throughout: "molecular cm-2" should be " molec.cm-2 "
**Author's Response:** Done.

iv) Abstract line 12: "ice in Kings Bay area, **which** emerged only …"
**Author's Response:** Done.

v) The quality of English needs checking throughout, e.g. "in consistency" is not an English phrase and should be replaced.
**Author's Response:** Revised.

vi) Introduction: " A typical heterogenous reaction model between gaseous and condensed phases **is** shown in Fig. 1"
**Author's Response:** Done.

vii) Introduction: "Bromine is released from salty ice surfaces" – but Fig 1 says "aerosol"
**Author's Response:**
The "aerosol" in Fig.1 represents the interface of sea salt surface and the air, where the heterogeneous processes releasing RHS occurred. It has been reported in many remote sensing observations and model researches that a large reservoir of halogen in the atmosphere is sea salt aerosol.

viii) Section 2.2 either use dSCD or DSCD but not both.
**Author's Response:** Done.

ix) Towards the end of Section 2.2 "much attention should be paid on the large elevation angles" – define what you mean by "large".
**Author's Response:** Since the inaccuracy of modeled BrO becomes larger at lower elevation angles, elevation angles of ≥8° should receive more attention.

x) Towards the end of Section 2.2: Change "From Fig.5b we can see obviously that the measured BrO DSCDS before midnight are in good consistence with…" to "From Fig.5b we can see obviously

that the measured BrO DSCDS before midnight are **best reproduced by**…"

**Author's Response:** Done.

xi) Towards the end of Section 2.2: "**This suggests that the** BrO layer between 0- 1km can be considered as the most **likely** distribution."

**Author's Response:** Done.

xii) Section 2.3: "According to the **radio**sonde records of…"

**Author's Response:** Done.

xiii) Section 2.3: "..height of the boundary layer is around 1200 meters at NyAlesund" – what is the range of boundary layer height, and is it possible to say what was it on 26th April at the start of the O3 loss/BrO increase?

**Author's Response:** From the backscatter coefficient data and the radiosonde records, the boundary layer height did not apparently change during the period.

xiv) Section 2.3: The trajectories shown are not "ensemble" trajectories – remove the word "ensemble" at the end of section 2.3.

**Author's Response:** Done.

xv) Section 4.1, first paragraph "Then we calculated the air mass backwards trajectory ending at 18:00 (UTC) 26th April in every hour (Fig 9**b**). i.e. not Fig 9a here.

**Author's Response:** Revised.

xvi) Section 4.2 – "The ice-sea water mixture was filled in the gaps, which was saltyenriched." – What evidence do you have that it was salty..??

**Author's Response:** Removed.

xvii) Section 4.2, second paragraph – if more than 80% of carbonate precipitates, will it make things acid, or only less alkali..?? Why should they become acid?

**Author's Response:** This sentence has been removed.

xviii) Section 4.2, second paragraph – "This process will provide acid aerosol from sea water" – do the authors really mean it will produce *aerosol*..?? If so, what is the mechanism..? What evidence do the authors have that the surface is airborne..? Throughout the majority of the paper they refer to sea-ice… Which surface is the one that matters..?

**Author's Response:** The sea salt aerosol above sea ice is where the heterogeneous reactions occurred. Here, aerosol refers to the interface of sea salt surface and the air. Please find the references in vii).

xix) Section 4.2 – If the authors still want to discuss influence of temperature with respect to ozone loss, they should refer to previous work looking at the link between these processes, e.g.

Tarasik and Bottenheim, ACP 12, 197, 2002. Note that Tarasik and Bottenheim suggest -20C is the temperature able to trigger ODEs.

**Author's Response:** The lowest temperature of Kings Bay at the night of 26 April is -11.4 $^{o}$C. According to the reference, it may not the direction cause of the ozone depletion. So we removed this sentence.

xx) Section 4.2 – again, the authors write "The sea ice is not totally fresh ice but the low air and water temperature in the fjord might cause the formation of brine ice mixture which is rich in sea salt aerosols" – the brine may be rich in sea salt, but aerosol only refers to sea salt once airborne – please correct this.

**Author's Response:** Revised.

xxi) Conclusions – again, the authors refer to "low temperature provide acid aerosols" – do they really mean that the surfaces are airborne? This needs clarification or correction

**Author's Response:** Revised.

xxii) Conclusions – statements about kinetics calculations need to be removed, as per Major concern described above.

**Author's Response:** Revised.

xxiii) Fig 2 b and c – maps are poor quality and need to be improved.

**Author's Response:** The maps are copied from the web site http://nsidc.org/soac. The quality is hard to improve. If it is not appropriate to put it here, we wonder if it can be put into the Appendix.

xxiv) Fig 5 – caption – The modelled DAMF (a) and BrO slant columns (b) – but (b) is now DSCD … Also, is fig c) SCD or DSCD?

**Author's Response:** Revised.

xxv) Fig 10 – quality is somewhat improved with the large images, but they are still hard to read. Please improve, and indicate location of Spitzbergen.

**Author's Response:** The location of the station in Spitzbergen has been marked.

xxvi) Fig 11 – needs information on source of photo, in particular to clarify that it is not the Kings Bay web cam.

**Author's Response:** Revised.

[revised manuscript text omitted]

---

## Author Response (AR4)

**Response to the Editor's comments**

Reviewer 3 suggested changes to the manuscript, that were needed to deal with the question whether or not the data can be used to demonstrate that ozone loss and bromine enhancements were "very fast". The authors initially argued that the changes were happening *in situ,* caused by local chemical processes, and were thus surprisingly fast. However, Reviewer 3 (supported by earlier comments from Reviewer 1) concluded that the reason the changes were fast was because they were driven by transport, something which has been reported previously in the literature. The authors made some requested changes. However, they still present their results on the basis that the rate of change is surprising – if it is transport-driven, the rates of change are not surprising, and should not be over-stated; indeed such changes and rates of change, have been reported in previous papers. Therefore amendments to the manuscript are still needed in a number of places, to adjust the tone of their paper to reflect this fact.

**Author's Response:**

We are very grateful for the reviewer and editor's valuable comments. Please kindly find the author's responses below.

There are two ways to address the major concerns raised. One would be the following:

i) Remove the sentence (**page 1 line 18 to 20**) that states: "the ozone loss rate during the bromine enhancement period was 10.3 ppbv $h_{-1}$, which is extremely high compared to those observed in other areas"

**Author's Response:** The abstract has been revised accordingly.

ii) On **page 2, line 32**, please change the statement " a unique process event" and replace with "an event"

**Author's Response:** Revised.

iii) **Page 8 line 4 and 5**, change the sentence: "The concurrent changes in the chemical and meteorological variables demonstrate the impact of environment change on this ozone depletion/BrO event" and replace with "The concurrent changes in the chemical and meteorological variables demonstrate that changes in observed chemistry are evident because of changes in transport, albeit on a small scale".

**Author's Response:** Revised.

iv) **Page 8 line 17 to 19**, please remove the first three sentences of this section; the section should therefore start "The deposition of gaseous mercury…."

**Author's Response:** Part 4.3 has been revised accordingly.

v) **Page 8 line 23**, remove the statement "The mercury loss rate is ~25 $ngm_{-3}h_{-1}$ or 6 $ngm_{-3}d_{-1}$

**Author's Response:** Removed.

vi) In the conclusions section**, Page 9 lines 2 to 7,** please remove all the text from "The concurrent changes in chemical and meteorological variables.......Further observations are required to identify its chemical mechanisms"

**Author's Response:** Removed.

vii) **Page 9 line2**, alter the text to read: "By analysing the air mass history and sea ice conditions, this BrO enhancement event was found to more likely be a regional process, driven by changes in sea ice and transport on a local scale."

**Author's Response:** Revised.

viii) **Table 2** should also be removed.

**Author's Response:** Done.

The alternative is that the authors ensure that every time they refer to the rate of changes calculated, they clearly state that these are most likely driven by transport, and then compare the calculated rates of change to those observed by others, and published in the literature. The context is extremely important.

I note also the Author's reply to the minor comment vii), regarding aerosol as surfaces for heterogenous reactions. The authors are correct, that sea salt aerosol has been shown to be an important source of bromine compounds to the atmosphere. However, they confuse the range of different surfaces which can act as a source. For example, in general, salty condensed phases can be a source of bromine compounds, and this can be achieved directly, and not only via producing aerosol. For example, there is evidence now that frost flowers are not a source of aerosol, however they could still be a modest source of bromine compounds directly to the atmosphere. To address this the authors should remove mention of aerosol on **page 2 lines 6 and 8**. On **page 6 line 25**, they should change wording to read: "...frost flowers, which can provide highly concentrated saline surfaces, and also sea salt aerosol." They should also remove the label "aerosol" in **Fig 1**. My view is that the discussion of bromine sources would then be consistent.

**Author's Response:** We agree with you and have revised the relevant parts following your suggestions.

[revised manuscript text omitted]